# Co-Culturing of Endothelial and Cancer Cells in a Nanofibrous Scaffold-Based Two-Layer System

**DOI:** 10.3390/ijms21114128

**Published:** 2020-06-10

**Authors:** Ye-Seul Oh, Min-Ho Choi, Jung-In Shin, Perry Ayn Mayson A. Maza, Jong-Young Kwak

**Affiliations:** 1Department of Pharmacology, School of Medicine, Ajou University, Suwon 16499, Korea; baram1825@ajou.ac.kr; 2Department of Biomedical Sciences, The Graduate School, Ajou University, Suwon 16499, Korea; minho8068@naver.com (M.-H.C.); Jungin@ajou.ac.kr (J.-I.S.); perryayn@ajou.ac.kr (P.A.M.A.M.); 3Immune Network Pioneer Research Center & 3D Immune System Imaging Core Center, Ajou University, Suwon 16499, Korea

**Keywords:** 3D cell co-culture, electrospinning, nanofibrous scaffold, endothelial cell, cancer cell

## Abstract

Angiogenesis is critical for local tumor growth. This study aimed to develop a three-dimensional two-layer co-culture system to investigate effects of cancer cells on the growth of endothelial cells (ECs). Poly(ε-caprolactone) (PCL) nanofibrous membranes were generated via electrospinning of PCL in chloroform (C-PCL-M) and chloroform and dimethylformamide (C/DMF-PCL-M). We assembled a two-layer co-culture system using C-PCL-M and C/DMF-PCL-M for EC growth in the upper layer with co-cultured cancer cells in the lower layer. In the absence of vascular endothelial growth factor (VEGF), growth of bEND.3 ECs decreased on C/DMF-PCL-M but not on C-PCL-M with time. Growth of bEND.3 cells on C/DMF-PCL-M was enhanced through co-culturing of CT26 cancer cells and enhanced growth of bEND.3 cells was abrogated with anti-VEGF antibodies and sorafenib. However, EA.hy926 ECs displayed steady growth and proliferation on C/DMF-PCL-M, and their growth was not further increased through co-culturing of cancer cells. Moreover, chemical hypoxia in CT26 cancer cells upon treatment with CoCl_2_ enhanced the growth of co-cultured bEND.3 cells in the two-layer system. Thus, EC growth on the nanofibrous scaffold is dependent on the types of ECs and composition of nanofibers and this co-culture system can be used to analyze EC growth induced by cancer cells.

## 1. Introduction

Neovasculogenesis occurs after endothelial cell (EC) proliferation and migration [1,2]. Angiogenesis is critical for local growth and metastasis of malignant tumors, and the tumor vasculature is structurally abnormal [3]. Tumors located > 100–200 μm from neighboring blood capillaries often encounter hypoxic conditions. Thus, hypoxia is one of the primary factors inducing tumor angiogenesis, thus upregulating vascular endothelial growth factor (VEGF) in tumor cells under hypoxia [4,5]. Exposure to hypoxia significantly upregulates hypoxia-inducible factor (HIF)-1α to transcriptionally regulate VEGF [6]. CoCl_2_ treatment of cells in vitro reportedly induces cellular changes similar to those observed upon hypoxia condition [7]. CoCl_2_ mimics hypoxia, at least in part, by occupying the Von Hippel–Lindau-binding domain of HIF-1α, thus preventing the degradation of HIF-1α [8], although there are certain differences between physiological hypoxia and CoCl_2_-induced hypoxia [9].

EC-tumor cell interaction in tissues occurs under three dimensional (3D) conditions. Furthermore, VEGF secretion depends on both a 3D environment and oxygen status [10]. A major limitation of conventional two-dimensional (2D) cell cultures in vitro is the failure to simultaneously culture two different cell types. Several 3D models to mimic angiogenesis in tumor tissue have been developed using various scaffolds and systems [11,12,13]. Most studies investigating in vitro EC-tumor cell interactions have used collagen and other hydrogels [14,15,16,17]. Furthermore, hypoxia promotes angiogenesis in ECs and adipose-derived stromal cells cultured in 3D gels [18]. However, few in vitro studies have focused on the direct effects of tumor cells on the growth of ECs under 3D conditions and the effects of hypoxic-inducing compounds on growth of ECs and cancer cells owing to the lack of a 3D co-culture system [19]. Moreover, chemical-induced hypoxia affects cell viability in a 2D culture system. Therefore, 3D co-cultivation is required to overcome the limitations including detection and cytotoxicity for long-term determination of the EC growth rate in co-culture with cancer cells.

Fabricated nanofibrous scaffolds have been developed for cell culture because nano-/micro-fibers have a similar spatial dimensionality to the fibrous component of the extracellular matrix (ECM), which comprises collagens 50–150 nm in diameter [20,21]. Electrospinning is a nanofiber fabrication method potentially yielding continuous fibers. Various synthetic polymers including poly(ε-caprolactone) (PCL), poly(D-lactide), and poly(vinyl alcohol) have been electrospun [22]. Electrospun PCL nanofibrous membranes (PCL-M) have been used as scaffolds to promote the growth, proliferation, and differentiation of various types of cells because they are highly biocompatible [23,24]. To enhance EC proliferation and function, numerous electrospun nanofibrous membranes from biocompatible and biodegradable synthetic polymers have been used [25]. The biocompatibility and biodegradability of PCL polymers have been demonstrated in biological applications [26,27]. Earlier reports showed that vascular graft of PCL-M induces endothelialization and ECM formation, accompanied by degradation of implanted PCL nanofibers [27]. However, cellular behavior and cell-specific growth of ECs on nanofibrous scaffolds have not been investigated. Furthermore, favorable scaffolds to detect differential EC growth in response to angiogenic factors secreted from cancer cells has not been tested. The broad range of fiber diameters from nano- to submicron-scale results in differences in fiber density, mean pore diameter, and pore distribution [28]. Thus, the topography of various nanofibrous scaffolds influences cellular behavior [29]. For example, fiber diameter affects cell adhesion and proliferation in PCL nanofibrous scaffolds [30]. For a particular polymer-solvent system, the choice of solvent influences the formation of PCL nanofibers significantly and is one of the factors influencing the production of various fibers and potentially influencing nanofiber properties [31,32]. Chloroform is the most commonly employed solvent for PCL electrospinning due to PCL solubility in chloroform [33]. *N*,*N*-Dimethylformamide (DMF) due to its higher electric conductivity and lower vapor pressure, was used as a solvent additive to generate PCL nanofibers [34,35]. It was shown earlier that the electrospun PCL nanofiber diameter decreases with an increasing proportion of DMF in dichloromethane and DMF mixed solvents [35].

In this study, to mimic tumor-driven angiogenesis, we produced nanofibrous membranes by electrospinning PCL dissolved in DMF and chloroform mixture and investigated the growth of ECs co-cultured with cancer cells in a 3D co-culture system, using a nanofiber-based two-layer system. Furthermore, hypoxia-mimicking conditions using CoCl_2_ were applied to this system to analyze hypoxia-induced EC growth co-cultured with cancer cells.

## 2. Results

### 2.1. Morphology of Electrospun C/DMF-PCL-M and C-PCL-M

PCL nanofibrous membranes were generated via electrospinning of PCL in chloroform (C-PCL-M) and chloroform and dimethylformamide (C/DMF-PCL-M) [28]. We investigated the effects of solvent on the morphology and size of electrospun fibers. Both C/DMF-PCL-M and C-PCL-M had an average thickness of 57.0 ± 10.2 μm (*n* = 15). The ultrastructure of nanofibrous membranes was analyzed via SEM. The nanofibers in both membranes were randomly oriented and structurally resembled collagen (Figure 1A). The structure of electrospun nanofibers showed a uniform distribution without bead formation. Most fibers in C/DMF-PCL-M had a diameter between 500 nm and 1.5 μm (0.97 ± 0.35 μm), whereas those of C-PCL-M had a diameter between 300 nm and 5 μm (3.86 ± 2.49 μm), indicating that C/DMF-PCL fibers had a narrower range of fiber diameter than C-PCL (Figure 1B). When the pore sizes for C/DMF-PCL-M and C-PCL-M were determined using ImageJ, C/DMF-PCL-M had a lower porosity than C-PCL-M. In a 1:1 chloroform:DMF mixture, the diameter of the fibers was between 300 and 750 nm (470 ± 70 nm) (data not shown). Thus, more uniform fibers and smaller pores formed in C/DMF-PCL-M than C-PCL-M because microfibers in C-PCL-M introduced larger pores than nanofibers.

### 2.2. Growth of ECs Seeded on C/DMF-PCL-M and C-PCL-M

The adhesion and spreading of ECs in a nanofibrous scaffold were evaluated after culturing ECs on the C/DMF-PCL-M and C-PCL-M without exogenous supplementation of VEGF in the culture media. In this study, bEND.3 mouse ECs and EA.hy926 human ECs were used. bEND.3 cells are immortalized cerebral microvascular ECs and exhibit the key features of ECs of the blood–brain barrier [36], whereas EA.hy926 cells are human umbilical vein cells established by fusing primary human umbilical vein cells with a thioguanine-resistant clone of A549 cells and have been used for in vitro studies on angiogenesis [37,38]. The cells exhibiting the morphological, phenotypic, and functional characteristics of mouse and human ECs were chosen for our study and have been used for studying the EC migration and formation of capillary-like tubules [39,40]. ECs were seeded onto the membranes for 1 d and then fixed to assess cellular adhesion. As shown in Figure 2A, bEND.3 cells and EA.hy926 cells adhered to the nanofibers and were well-distributed throughout the scaffold in both nanofibrous membranes 1 d after seeding. Thus, cellular adhesion to C/DMF-PCL-M and C-PCL-M did not significantly differ between bEND.3 and EA.hy926 cells. The tight junction adaptor protein zona occludin (ZO)-1 is essential for barrier formation in microvascular EC and regulates the migration and angiogenic potential of ECs [41]. The density of phalloidin- and ZO-1-labeled bEND.3 cells exhibiting green and red fluorescences in the C/DMF-PCL-M significantly decreased 3 d after culturing. In comparison to C/DMF-PCL-M, the growth of bEND.3 cells on C-PCL-M was stable. However, the fluorescence intensity of EA.hy926 cells on both C/DMF-PCL-M and C-PCL-M increased after 3 d of culturing. At 5 d after culturing, EA.hy926 cells, but not bEND.3 cells, on C/DMF-PCL-M retained their morphology in the scaffold. SEM revealed that bEND.3 and EA.hy926 cells cultured for 1 d in the scaffold adhered and spread well along the nanofibers, displaying distinct morphologies on the scaffold surfaces (Figure 2B). With time, the morphology of bEND.3 cells in C/DMF-PCL-M was changed from an elongated form to a spherical form. In contrast, bEND.3 cells on C-PCL-M and EA.hy926 cells on both nanofibrous membranes exhibited a more extended morphology rather than an ovoid morphology after 5 d of culturing. Similarly, a previous study reported that human coronary artery ECs cultured on C/DMF-PCL-M retained a spherical morphology from the beginning of cell seeding and did not spread with time [42]. Together, these results suggest that EC growth on the nanofibrous membrane depends on cell type and the composition, structure, and distribution of nanofibers.

### 2.3. The Growth of ECs Co-Cultured with Cancer Cells in a Nanofibrous Membrane-Based Two-Layer Culture System

Hypervasculature involvement in metastatic colorectal cancer and hepatocellular carcinoma development was shown earlier [43,44]. We used CT26 murine colon cancer and HepG2 human hepatocellular cancer cells since they secrete VEGF and induce angiogenesis [45,46]. We assessed whether the growth of cancer cells is different on C/DMF-PCL-M and C-PCL-M. The degree of cell distribution increased in both conditions (Figure 3A), however, the proliferation rate was higher on C-PCL-M than on C/DMF-PCL-M (Figure 3B). Thus, in this study, cancer cells were cultured on C-PCL-M and not on C/DMF-PCL-M. In addition, the large pores in the C-PCL-M might facilitate deeper infiltration of cells into the scaffold and help maintain cell aggregates in the scaffold [28].

To examine the effect of cancer cells on EC growth, we set up the two-layer co-culture system using ECs and cancer cells cultured on the upper and lower nanofibrous membranes, respectively (Appendix A). With time, cell morphology and distribution clearly differed in the upper C/DMF-PCL-M layer between the monoculture of bEND.3 cells and co-culture with CT26 cancer cells (Figure 4A). Immunofluorescence analysis of the upper C/DMF-PCL-M layer revealed a reduction in the number of fluorescent ECs in the monoculture, however, the fluorescence intensity of bEND.3 cells increased upon co-culturing with CT26 cancer cells. In a focus-stacking analysis of confocal microscopy images obtained at 5 μm intervals, fluorescently labeled bEND.3 cells were detected in the scaffold at a depth of 25–35 μm on day 1 after culturing to 10–25 μm on day 5 with loss of phalloidin staining without co-culturing with CT26 cells (Figure 4B). However, in co-culture conditions, most ECs were detected in deeper layers from the upper surface on day 5 rather than on day 1 with an increased density of phalloidin staining. When the total intensity of red fluorescence in the upper layer was examined using ImageJ, the fluorescence intensity was higher among ECs co-cultured with cancer cells on C/DMF-PCL-M rather than in EC monocultures (Figure 4C). However, notwithstanding an increase in the fluorescence intensity on C-PCL-M in the EC monoculture alone and the co-culture, no significant difference was observed between the two conditions. The CCK-8 assay revealed that the number of viable bEND.3 cells on the C/DMF-PCL-M slightly decreased from day 1 to day 5 without CT26 cells, however, it was significantly increased upon co-culturing with CT26 cells (Figure 4D). The cell proliferation rate of bEND.3 cells on C-PCL-M did not display a significant difference between the EC monoculture and co-culture. Furthermore, the morphology and distribution of EA.hy926 cells on both nanofibrous membranes were not prominently changed upon co-culturing with HepG2 cells on the lower C-PCL-M membrane (Figure 5A). The proliferation rate of EA.hy926 cells on the upper layers increased at 5 d after culturing alone and did not further increase upon co-culturing with HepG2 cells (Figure 5B).

### 2.4. Growth Regulation of bEND.3 Cells on C/DMF-PCL-M by VEGF Produced from Co-Cultured Cancer Cells

We investigated whether VEGF induces bEND.3 cell growth upon co-culturing with cancer cells because EC growth was significantly decreased without exogenous VEGF and recovered upon co-culturing with cancer cells. bEND.3 cells are known to express VEGF receptor-1 and VEGF receptor-2 [47]. First, experiments were performed with bEND.3 cells cultured on C/DMF-PCL-M with VEGF at different concentrations. As shown in Figure 6A, VEGF increased the distribution of fluorescently labeled cells in a dose-dependent manner for 5 d in culture. The cell proliferation assay revealed an increased proliferation of bEND.3 cells by VEGF (Figure 6B). Moreover, the VEGF-induced increase in EC growth and proliferation was blocked significantly upon treatment with anti-VEGF antibodies. These results suggest that VEGF induced growth and proliferation of bEND.3 cells in a 3D culture condition. In this study, EA.hy926 cell growth was maintained in the 3D culture, whereas the growth of primary human umbilical vein endothelial cells (HUVECs) was not significantly increased in the presence or absence of VEGF (data not shown). Reports show that EA.hy926 cells express both VEGF receptor-1 and VEGF receptor-2 [48,49]. Upon VEGF treatment of Ea.hy926 cells seeded on C/DMF-PCL-M, the cells displayed an elongated morphology and increased growth after short-term treatment. However, the effect of VEGF on EA.hy926 cell growth was less significant after prolonged culturing (Appendix A). The formation of tight junctions between adjacent ECs was clearly observed in the 2D culture (Figure 7A), whereas ZO-1 expression was primarily detected throughout the cytoplasm of the ECs cultured in the nanofibrous scaffold. In a focus-stacking analysis of confocal microscopy images of ZO-1 expression, VEGF-treated bEND.3 cells maintained an elongated cell morphology and a few cell-cell junctions were observed in C/DMF-PCL-M (Figure 7B). In addition, the growth and tight junction formation of bEND.3 ECs cultured in C-PCL-M were not significantly increased by VEGF treatment. Patterns of ZO-1 expression were similar to that of bEND.3 cells in EA.hy926 ECs cultured in culture plates and PCL-M (Appendix A). The lower formation of tight junctions in the porous nanofibrous scaffold than on the 2D culture plates may be due to the adhesion and growth of ECs along nanofibers rather than the formation of tight junctions on flat surfaces.

We assessed the levels of VEGF secreted from cancer cells cultured on C-PCL-M. VEGF secretion by CT26 cells increased in a time-dependent manner in both 2D culture plates and 3D nanofibrous membranes (Appendix A). HepG2 cells cultured on C-PCL-M secreted more VEGF than CT26 cells (Appendix A), although co-culture of HepG2 cells failed to increase EA.hy926 cell growth compared to the culture of EA.hy926 cell alone. Thereafter, when anti-VEGF antibodies were supplemented in co-cultures of bEND.3 and CT26 cells, bEND.3 cell growth was significantly decreased in a dose-dependent manner (Figure 8A). As shown in Figure 8B, the proliferation of bEND.3 cells upon co-culturing with CT26 cells was significantly inhibited by anti-VEGF antibody. Sorafenib inhibits the VEGF receptor signaling cascade, thereby blocking tumor angiogenesis [50]. Herein, the growth of bEND.3 cells was enhanced upon co-culturing with CT26 cells and upon exogenous supplementation of VEGF, but EC growth was significantly inhibited upon treatment with 1 μM sorafenib (Figure 8C). Together, EC growth was maintained by cancer cells in a VEGF-dependent manner in the 3D co-culture system, and VEGF is the most potent factor for inducing EC growth, although we cannot rule out the presence of growth-enhancing factor(s) other than VEGF secreted from cancer cells. Moreover, using the present culture system, we investigated the efficacy of the VEGF receptor antagonist on the growth of ECs induced by cancer cells.

### 2.5. Effects of CoCl_2_ on Cell Growth, HIF-1α Expression, and VEGF Production in Cancer Cells Cultured on C-PCL-M

Tumor hypoxia is an important stimulus causing increased VEGF production [51]. We further determined whether hypoxia-mimicking conditions using CoCl_2_ affected EC growth upon co-culturing with cancer cells. Upon supplementation of CoCl_2_ (100 μM) to the co-culture of bEND.3 ECs in the upper layer and CT26 cells in the lower layer, CoCl_2_ did not influence EC growth up to 5 d of culturing (data not shown). By comparing the cell distributions and densities of fluorescently labeled cells on the C-PCL-M layer, CT26 cells in monoculture condition displayed a significant reduction in cell numbers and growth upon treatment with 100 μM CoCl_2_ for 3 d in comparison with untreated control cells (Figure 9A). HIF-1α is an essential regulatory factor facilitating cellular adaptation to hypoxia [52]. Treatment of CT26 cells with CoCl_2_ at different concentrations for 5 d revealed that 150 μM CoCl_2_ significantly upregulated HIF-1α in surviving cancer cells. When proliferation rates were compared using CCK-8 assay from day 1 to day 5, the number of viable CT26 cells decreased in a time-dependent manner upon treatment with CoCl_2_ at different concentrations (Figure 9B). In parallel, we compared the effects of CoCl_2_ on the growth and proliferation of cancer cells between the 2D culture plate and 3D nanofibrous membrane. The inhibition of CT26 cell proliferation by CoCl_2_ on the C-PCL-M was lower than that on the culture plate (Appendix A). Furthermore, we determined the amounts of VEGF secreted from CoCl_2_-treated cancer cells. Despite a slight elevation in VEGF, no significant difference was observed between untreated and CoCl_2_-treated conditions (Figure 9C). As CoCl_2_ suppressed cancer cell proliferation, we confirmed these results by normalizing VEGF concentrations to CCK-8 levels. As shown in Figure 9D, the ratio of VEGF concentrations to CCK-8 levels was higher in CoCl_2_-treated cells than in untreated cells because surviving CT26 cells at lower numbers after CoCl_2_ treatment might have produced more VEGF than untreated-cells at higher numbers.

### 2.6. Effects of CoCl_2_ on the Growth of ECs in the Co-Culture

We investigated the effects of hypoxia on EC growth in the co-culture. Decreased growth of bEND.3 cells after prolonged culturing was reversed upon co-culturing with cancer cells, however, EC growth was not further increased upon CoCl_2_ treatment in the co-culture. Furthermore, treatment of co-cultures with CoCl_2_ did not increase VEGF secretion from cultured CT26 cells. Thus, as shown in Figure 10A, the effects of CoCl_2_ on the growth of bEND.3 and CT26 cells were assessed in a state of restrained growth of ECs under modified co-culture conditions. bEND.3 cells were cultured on C/DMF-PCL-M for 2 d and CT26 cells were cultured on C-PCL-M cultured for 4 h for membrane adherence in each well. These two cell-membrane constructs were assembled as a two-layer system in the Transwell chambers and ECs and cancer cells were co-cultured in the presence or absence of CoCl_2_ (150 μM) for another 1 d. When bEND.3 cells were treated with CoCl_2_ in the presence of CT26 cells, the growth of bEND.3 cells was more prominent upon confocal microscopic analysis of the distribution and densities of labeled cells on the upper C/DMF-PCL-M (Figure 10B). After 1 d of treatment with CoCl_2_, the numbers of fluorescently labeled CT26 cells did not decrease, although HIF-1α upregulation was observed in CoCl_2_-treated CT26 cancer cells rather than in untreated cells. The CCK-8 assay in the upper layer containing ECs revealed a significant increase in the number of live bEND.3 cells upon co-culturing with CT26 cells and further increased upon addition of CoCl_2_ to the co-culture (Figure 10C). In comparison, the number of live CT26 cancer cells did not significantly decrease upon addition of CoCl_2_ to monoculture of CT26 cells and co-culture and did not increase upon co-culturing with bEND.3 cells, when compared to the monoculture of cancer cells (Figure 10D). Finally, we assessed VEGF secretion via ELISA. As shown in Figure 10E, the concentrations of VEGF secreted from CoCl_2_-treated CT26 cells for 1 d were significantly higher than those from untreated cells. VEGF concentrations were lower in co-cultures of CT26 and bEND.3 cells than in monoculture of CT26 cells. These lower VEGF levels in the media of co-cultures may result from the binding of secreted VEGF to cell membrane receptors in ECs. Together, the present results indicate that chemically induced hypoxia increased VEGF production in cancer cells and enhanced the growth of ECs in the nanofiber-based two-layer co-culture system.

## 3. Discussion

Angiogenic factors secreted by cancer cells modulate proliferation and survival of ECs [1], however, the tumor vasculature is abnormal and dysfunctional [53]. The tube formation assay is a standard in vitro method for assessing angiogenesis [54]. However, this method has a general limitation in in vitro studies related to the effects of co-cultured cancer cells on ECs to investigate tumor angiogenesis. Herein, bEND.3 cells exhibited spherical morphology and reduced growth on C/DMF-PCL-M in the absence of VEGF. These culture conditions indicate that bEND.3 cell growth critically depends on angiogenic factors including VEGF. Our co-culture model using bEND.3 cells and cancer cells on nanofibrous membranes in a two-layer system is a simplified 3D tumor angiogenesis model and provides valuable insights into the cross-talk between ECs and cancer cells. Although no direct interaction was observed between ECs and cancer cells in this co-culture system, the effects of cancer cells on EC growth are associated with the release of paracrine effectors, such as VEGF, rather than direct interaction between ECs and cancer cells. Hypoxic cancer cells locate more than 100 μm away from blood vessels and angiogenic factors produced by cancer cells bind to EC receptors [55]. Accordingly, the two-layer system specifically focuses on EC growth based on secreted factors from co-cultured cancer cells. Growth factors including VEGF released from cancer cells form and retain a chemical gradient within a 3D scaffold. Thus, indirect co-culturing of ECs and cancer cells may be beneficial for assays on tumor-induced angiogenesis. Moreover, as observed herein, the 3D in vitro system facilitates long-term co-culturing of ECs and cancer cells.

Several proteins activate EC growth and movement [56]. Herein, EC growth was assessed in VEGF-free media using a nanofibrous membrane-based 3D culture system. The growth of bEND.3 cells in VEGF-free media was significantly decreased on the C/DMF-PCL-M rather than on C-PCL-M. Moreover, the growth of bEND.3 cells on C/DMF-PCL-M was highly responsive to VEGF secreted from cancer cells. Upon VEGF stimulation and co-culturing with cancer cells, numerous elongated ECs were observed. These results suggest that EC growth in 3D co-cultures is significantly dependent on VEGF secreted from cancer cells. Hence, future studies are required to analyze the growth of ECs co-cultured with cancer cells in the absence of VEGF and other growth factors, using a specific 3D scaffold such as C/DMF-PCL-M. Moreover, VEGF has been rendered a potential therapeutic target in numerous solid malignant tumors. ECs co-cultured with cancer cells failed to grow and proliferate in the presence of sorafenib and anti-VEGF antibodies. Thus, the nanofiber-based co-culture system provides a suitable platform to investigate the effects of novel anti-angiogenesis agents on the growth of ECs in response to tumor-derived growth factors. Finally, stromal cells including fibroblasts and immune cells can be incorporated into tumor cells in the lower layer of the present system to investigate the effects of the microenvironment on tumor angiogenesis.

Hypoxia upregulates VEGF, leading to VEGF-dependent tumor angiogenesis [57]. VEGF secretion reportedly depends on both the 3D environment and oxygen status [10]. Herein, hypoxia-like conditions were chemically generated through treatment of cells with CoCl_2_ [8]. CoCl_2_ has been extensively studied for its therapeutic potential to stimulate cellular function regulated by HIF-1 [58]. This study shows that treatment of cancer cells with CoCl_2_ on culture plates resulted in a more pronounced reduction in cell proliferation compared to culturing on nanofibrous membranes. We thus generated an in vitro hypoxia-like condition via treatment of CT26 cancer cells with CoCl_2_ in 3D culture conditions, however, CoCl_2_ still inhibited cancer cell proliferation. In contrast to our expected observations, VEGF release from CT26 cancer cells was not significantly enhanced with high concentrations of CoCl_2_ after prolonged culturing owing to inhibition of cell proliferation. To overcome this issue in the assay, we used a two-layer co-culture system wherein ECs were cultured for 2 d in the absence of VEGF, and cancer cells were treated with CoCl_2_ for 1 d. Under this condition with CoCl_2_ treatment in the two-layer co-culture system, cancer cell growth was stable, HIF-1α was upregulated in surviving cancer cells, VEGF was produced at high levels, and EC growth was enhanced.

bEND.3 ECs displayed a different growth pattern and response to VEGF on C/DMF-PCL-M compared to the 2D culture. ECs were oriented three-dimensionally, and the contact surfaces between adjacent ECs were formed in the C/DMF-PCL nanofibrous scaffold, although the amount of tight junctions formed in the nanofibrous scaffold was low compared to that of the 2D culture. Differences were observed in the growth rates of ECs between 3D nanofibrous membranes and 2D culture plates. Furthermore, the growth and proliferation of bEND.3 ECs in the presence of VEGF were more prominent on C/DMF-PCL-M than on 2D culture plates. Proliferating ECs in the 2D culture conditions retained physical interactions with one another, resulting in a state of contact inhibition and a low proliferative response to prolonged VEGF stimulation [59]. Thus, it is difficult to assess the angiogenic effects of VEGF and other proangiogenic factors on highly proliferating cells in 2D or 2D-like culture conditions. However, ECs in the 3D culture conditions were sparse or not completely surrounded by adjacent cells. Hence, non-confluent ECs may have markedly responded to VEGF rather than their confluent counterparts and the responsiveness of ECs to VEGF and proangiogenic factors derived from co-cultured cancer cells seemed higher in 3D than in 2D culture conditions.

Micro/nanofibers were generated via electrospinning and electrospun PCL scaffolds were used for 3D culturing of cells [28]. The nanofiber structure of electrospun scaffolds reportedly influences cellular behavior including cell adhesion, proliferation, and differentiation, although the information regarding the mechanisms underlying these effects of the scaffolds on cellular behavior is limited [20,60,61]. The ECs on the nanofibers proliferated in specific patterns and formed a continuous monolayer [62]. Cell adhesion and proliferation are greater on nanofibers than on microfibers [20,61]. However, herein, C/DMF-PCL nanofiber scaffolds with smaller diameters and pores negatively affected bEND.3 cell growth in comparison with C-PCL nanofiber scaffolds with larger diameters and pores. ECs seeded on nanofibrous membranes with small pore sizes could not infiltrate the scaffold but rather adhered and spread on the surface of the polymer nanofibers [62,63]. Another study reported that electrospun microfibers with nanoporous features displayed increased spreading of human mesenchymal stem cells in comparison with smooth fibers, while maintaining the same pore size [64]. C/DMF-PCL-M and C-PCL-M were generated via electrospinning of PCL polymer solutions using different solvents. Thus, it may be speculated that solution composites constitute a factor underlying different growth patterns of ECs between the two membranes. DMF has a high dielectric constant and dipole moment, enhancing the conductivity of polymer solutions [34]. Thus, a more conductive polymer solution of C/DMF-PCL rather than C-PCL may harbor a greater electric charge during electrospinning, thus facilitating the generation of uniform fibers [34]. Furthermore, DMF in C/DMF-PCL-M might have inhibited the growth of cells owing to cytotoxicity, however, this possibility can be ruled out, as the growth of cancer cells was not inhibited on C/DMF-PCL-M and the reduction of the growth of bEND.3 cells was reversed by VEGF. PCL polymers do not harbor natural cell binding sites, and serum proteins adsorbed onto the polymer surface mediate cell adhesion to PCL nanofibers. Moreover, it is possible that cell binding ECM proteins, such as fibronectin can be produced by ECs and cancer cells and adsorbed to PCL nanofibers with different coating efficiency. Thus, differential adsorption of serum or ECM proteins in C/DMF-PCL and C-PCL nanofibers may affect EC binding to the scaffolds. Together, the chemical and physical properties of PCL nanofibers may affect EC growth in the absence of angiogenic factors.

In conclusion, the cross-talk between ECs and cancer cells can be investigated in nanofiber-based two-layer co-culture systems. A few bEND.3 cells were detected in the upper C/DMF-PCL-M layer when cancer cells were not seeded in the lower C-PCL-M layer, however, co-culturing with cancer cells enhanced the growth of bEND.3 cells. The growth of bEND.3 cells co-cultured with cancer cells on C/DMF-PCL-M was abrogated by sorafenib. Furthermore, supplementation of CoCl_2_ in the co-culture upregulated HIF-1α and increased VEGF production in cancer cells and EC growth. Together, the present model accounts for 3D growth of the specific types of ECs on nanofibrous membranes, wherein fibers have characteristic composition and structure. This system provides a suitable example showing the advantages of morphological analysis of the co-culture model and analysis of the effects of anti-angiogenic drugs on the interaction between ECs and cancer cells.

## 4. Materials and Methods

### 4.1. Materials

PCL (*M*_n_ = 700,000–900,000), chloroform, DMF, 4-(2-hydroxyethyl)-1-piperazineethanesulfonic acid (HEPES), dimethyl sulfoxide, CoCl_2_, and 4′,6-diamidino-2-phenylindole dihydrochloride (DAPI) were purchased from Sigma-Aldrich (St. Louis, MO, USA). Dulbecco’s modified Eagle’s medium (DMEM), RPMI-1640 medium, fetal bovine serum (FBS), penicillin/streptomycin, glutamine, and 0.05% trypsin-ethylenediaminetetraacetic acid were purchased from Gibco (Rockville, MD, USA). Polydimethylsiloxane (PDMS) was purchased from Dow-Corning Korea, Inc. (Seoul, Korea).

### 4.2. Electrospinning and Fabrication of PCL Nanofibers

Porous PCL nanofibers were generated in accordance with a previously reported method [28,65]. Briefly, the polymer for electrospinning was dissolved in 99.5% pure chloroform and a chloroform:DMF mixture (3:1, *v/v*) at 15% and stirred at 37 °C for 5 h to obtain a homogeneous solution. Nanofiber membranes were fabricated via electrospinning (NanoNC, Seoul, Korea). Two-nozzle spinnerets were used with an average flow rate of approximately 8 μL/min, using a syringe pump. Each nozzle had an inner diameter of 210 μm (27 G). Nanofibers were collected onto a rotating metallic mandrel at 100 rpm at ambient temperature for 4 h. The nozzle tip-to-collector distance was set at 20 cm, with an electrical potential of approximately 17.5 kV from the grounded collector plate. Membrane thickness was measured using an ultraprecision micrometer (Mitsutoyo Co., Kawasaki, Japan). The morphology of electrospun fibers was observed using scanning electron microscopy (SEM) with a model SEM4500 (Sec, Suwon, Korea). Fiber diameter and pore size were measured by an average of 20 random measurements from SEM images of five membranes using the ImageJ software (ImageJ, National Institutes of Health, Bethesda, MD, USA). The electrospun membranes were sterilized by soaking in a solution of 70% ethanol for 12 h and dried under UV exposure for 12 h.

### 4.3. Cell Culture

bEND.3 and EA.hy926 ECs were purchased from American Type Culture Collection (ATCC) (Manassas, VA, USA) and cultured in ATCC-formulated DMEM supplemented with 10% FBS, 1% penicillin/streptomycin, 20 mM HEPES, and 2 mM L-glutamine at 37 °C in 5% CO_2_. ECs were sub-cultured every 3–4 d. Culture media was changed after 24 h of sub-culturing and every 2 d thereafter. ECs at passages 6–10 were used herein. CT26 murine colon carcinoma cells, syngeneic to BALB/c mice, and HepG2, human hepatoma cells, were obtained from Korean Cell Bank (Seoul, Korea) and grown as monolayer cultures in DMEM containing 10% FBS supplemented with penicillin/streptomycin.

### 4.4. Cell Culture in a Nanofibrous Membrane

PDMS (100 μL) was poured in 8-well plates and kept on a slide warmer at 100 °C for 3 min. The PCL-M (1 cm × 1 cm) was attached to the surface of gel-state PDMS on 8-well plates and immersed in DMEM (700 μL) for 6 h at 37 °C to increase cell adhesion. ECs and cancer cells were seeded on the nanofibrous membrane at 30,000 cells/membrane. After 4 h of cell adhesion, ECs and cancer cells on the membranes were cultured in DMEM (700 μL) containing 10% FBS supplemented with penicillin/streptomycin for up to 5 d. For the 2D culture system, the same procedures were followed on an 8-well plate.

### 4.5. Co-Culture of ECs and Cancer Cells in a Nanofibrous Membrane-Based Two-Layer System

We assembled a two-layer system using PCL-M for EC growth with co-cultured cancer cells, similar to a modified Transwell chamber (Appendix A). To generate the upper layer in the apical chamber, the polycarbonate filter of a Transwell insert was removed and replaced with a 5-mm-diameter circular section of C/DMF-PCL-M and C-PCL-M, which was attached with gel-state PDMS (40 μL). For the lower layer, PDMS solution was placed in a 24-well plate and the volume of PDMS was adjusted to 220 μL to maintain a constant level of gels in the lower compartments and constant distance from upper surface of the lower membrane to the lower surface of the upper membrane. Thereafter, a C-PCL-M was attached on top of the surface of the gel-state PDMS in the lower well. EC and cancer cell suspensions (10 μL) were seeded at 30,000 cells/well on the membranes in the upper and lower layers, respectively and incubated for 4 h for adherence to the membranes. The two layers were then assembled layer-by-layer by placing the insert in the well, and cells were co-cultured in DMEM containing 10% FBS without VEGF for up to 5 d. In the two-layer system, bEND.3 cells were co-cultured with CT26 cells and EA.hy926 cells were co-cultured with HepG2 cells at 37 °C in a humidified CO_2_ atmosphere, respectively. At 1, 3, and 5 d post-seeding, the cell and nanofibrous membrane constructs were removed for morphological assessment.

### 4.6. Laser Confocal Microscopy

ECs and cancer cells cultured on culture plates and nanofibrous membranes were fixed with 4% paraformaldehyde in PBS for 10 min and permeabilized with 0.2% Triton-X-100 for 3 min at ambient temperature. After washing with PBS (3 × 10 min), the cells were blocked using 5% bovine serum albumin (BSA) in PBS for 1 h at ambient temperature and incubated overnight at 4 °C with rabbit polyclonal anti-ZO-1 antibody (1:50 dilution) (Invitrogen, Carlsbad, CA, USA), anti-HIF-1α antibody (1:50 dilution) (Abcam, Cambridge, UK), and anti-vimentin antibody (1:50 dilution) (Invitrogen). The samples were washed in PBS (3 × 10 min) and then treated with donkey anti-rabbit AlexaFluor 594 antibody (1:500 dilution) (Thermo Fisher scientific, Waltham, MA, USA) for 1 h at ambient temperature. The nuclei were counterstained with DAPI (1:500 dilution) for 10 min at ambient temperature and F-actin was labeled with AlexaFluor 488-conjugated phalloidin (1:400 dilution) (Sigma) for 1 h at ambient temperature. The labeled cells were visualized using a laser scanning confocal microscope (Nanoscope, Daejeon, Korea) at the 3D immune system imaging core facility of Ajou University. Images was analyzed using the ImageJ software.

### 4.7. SEM

Cells cultured on the membranes were rinsed twice with PBS and fixed in 2.5% glutaraldehyde in 0.1 M phosphate buffer for 24 h at 4 °C. Thereafter, the samples were immersed for 1 h in 1% OsO_4_ (Sigma) in 0.1 M phosphate buffer and were dehydrated with increasing concentrations of ethanol (50%, 70%, and 100%). All samples were affixed to aluminum mounts with double-sided carbon tape and coated with gold-sputter. Cellular morphology was observed using a SEM.

### 4.8. Cell Proliferation Assay

Cell growth on PCL-M was assessed using the cell counting kit (Cell Counting Kit-8 [CCK-8]; Dojindo Molecular Technologies, Gaithersburg, MD, USA) in accordance with the manufacturer’s protocol. Briefly, cells were seeded at 30,000 cells/mL in an 8-well plate and a nanofibrous membrane was attached to the 8-well plate. After 1, 3, and 5 d, 10% (*v/v*) CCK-8 solution in DMEM was added to each well, and cells were incubated for 2 h. When ECs and cancer cells were co-cultured in a two-layer system using a Transwell chamber, the upper layer of the Transwell insert was separated from the lower chamber culturing cancer cells and replaced in a new chamber, and CCK-8 solution was then added to each well containing ECs or cancer cells. Cell proliferation was investigated by measuring the absorbance at 450 nm, using a microplate reader Synergy H1 (Biotek, Seoul, Korea).

### 4.9. Cytokine Assay

VEGF secretion was analyzed using an ELISA assay Kit (R&D Systems, Minneapolis, MN, USA) in accordance with the manufacturer’s protocol. Cancer cells (3 × 10^4^) were cultured in the presence or absence of CoCl_2_ on the culture plates and nanofibrous membranes. After culturing for the indicated periods, supernatants were harvested and assayed.

### 4.10. Statistical Analysis

The results are presented as means ± standard deviation (SD) values. Student’s *t*-test was used to compare the means of paired or unpaired samples. A *p*-value of <0.05 was considered significant.

## Figures and Tables

**Figure 1 ijms-21-04128-f001:**
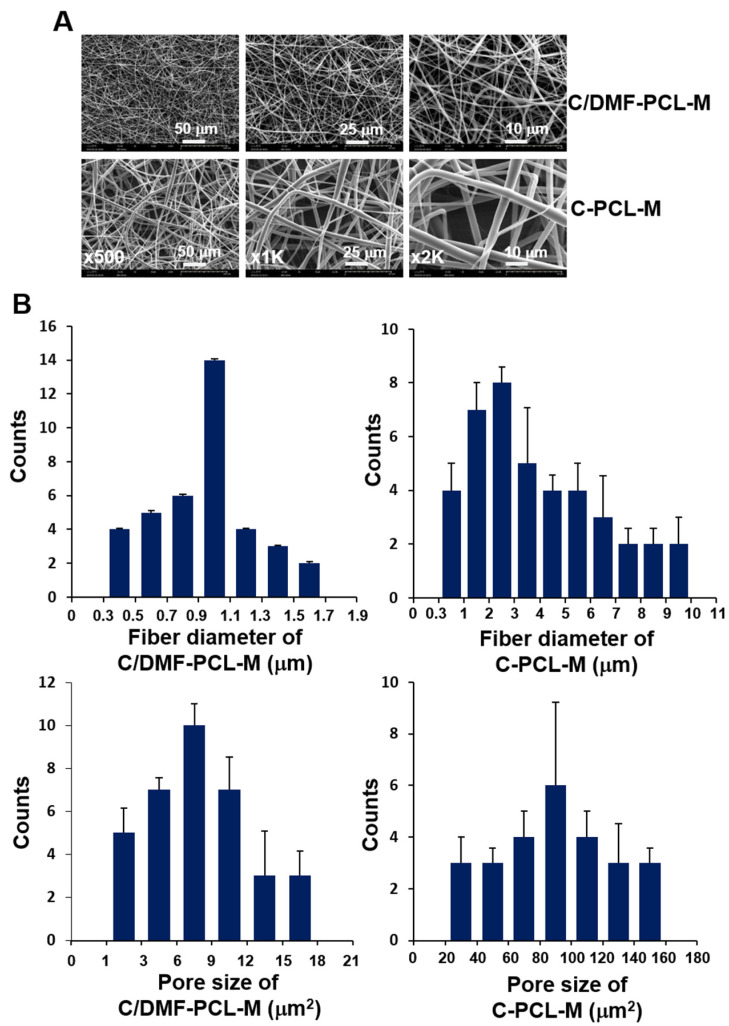
Fiber diameter and pore size distribution of electrospun Poly(ε-caprolactone) (PCL) in chloroform (C-PCL-M) and chloroform and DMF (C/DMF-PCL-M). (**A**) Fiber morphology in C/DMF-PCL-M and C-PCL-M was assessed via SEM. The results represent five independent experiments. (**B**) The frequency of fiber diameters and pore sizes in nanofibrous scaffolds was analyzed using ImageJ. Data are shown as mean ± SD values (*n* = 20).

**Figure 2 ijms-21-04128-f002:**
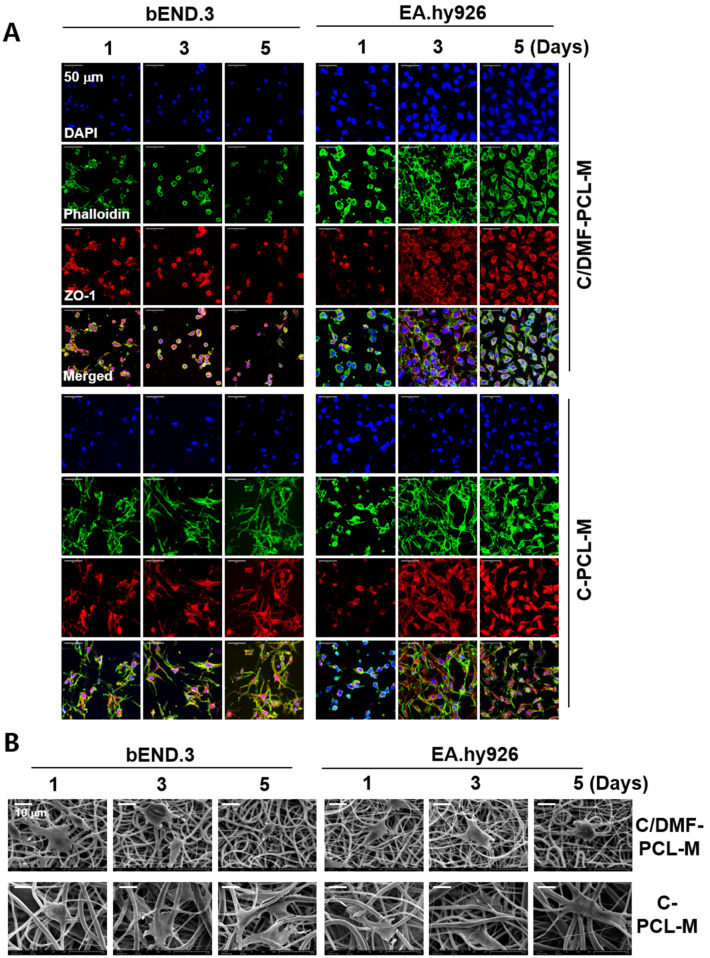
Different growth patterns of ECs on the C/DMF-PCL-M and C-PCL-M. bEND.3 and EA.hy926 ECs (3 × 10^4^) were top-seeded on C/DMF-PCL-M and C-PCL-M and cultured for the indicated periods (*n* = 4). (**A**) The growth and morphology of the cells were evaluated via confocal microscopy after cells were stained using DAPI (blue), phalloidin (green), and anti-ZO-1 antibody (red). (**B**) EC adhesion and morphology on nanofibrous membranes were evaluated via SEM. Data represent four independent experiments.

**Figure 3 ijms-21-04128-f003:**
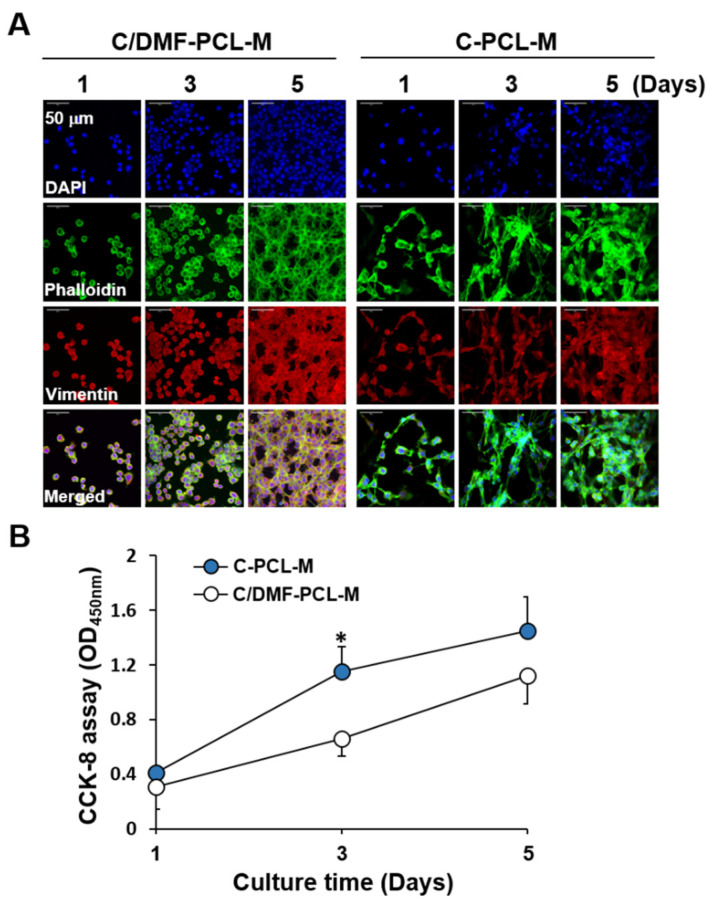
The growth of CT26 cancer cells on C/DMF-PCL-M and C-PCL-M. CT26 cells (3 × 10^4^) were top-seeded on the C/DMF-PCL-M and C-PCL-M and cultured for the indicated periods (*n* = 3). (**A**) Cell morphology was evaluated via confocal microscopy after immunofluorescence staining of the cells with DAPI (blue), phalloidin (green), and anti-vimentin antibody (red). (**B**) Proliferation rates of the CT26 cells were analyzed using the CCK-8 assay. Data represent three independent experiments. * *p* < 0.05, compared to C/DMF-PCL-M.

**Figure 4 ijms-21-04128-f004:**
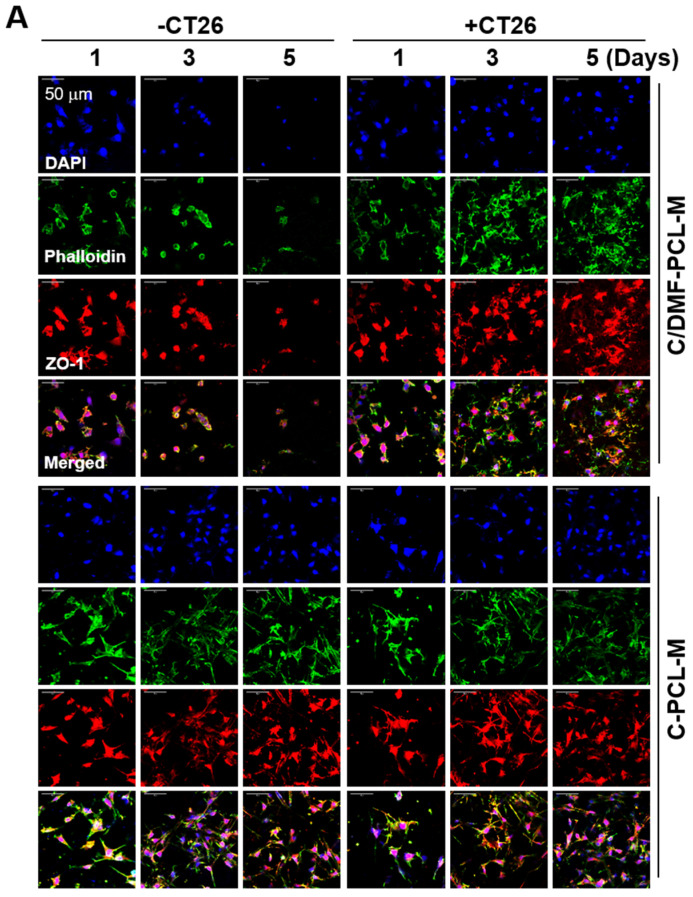
Enhanced growth of bEND.3 ECs co-cultured with CT26 cancer cells on C/DMF-PCL-M in a two-layer culture system. bEND.3 cells (3 × 10^4^) were top-seeded on either C/DMF-PCL-M or C-PCL-M in the upper layer and cultured with (+CT26) or without (−CT26) CT26 cells (3 × 10^4^) on the C-PCL-M in the lower layer as shown in the Appendix A. Cells were cultured for the indicated periods (*n* = 3) and stained with fluorescently labeled antibodies. (**A**) Cell distribution and morphology were assessed via confocal microscopy. Images represent three fields in each membrane. (**B**) Cellular localization in the nanofibrous scaffold was evaluated by focus-stacking images of the cell-scaffold construct (*n* = 3). (**C**) The intensity of red fluorescence was analyzed using ImageJ software. (**D**) Proliferation rates of the cells were analyzed using the CCK-8 assay. Data are presented as mean ± SD values (*n* = 3).* *p* < 0.05, compared to d 1.

**Figure 5 ijms-21-04128-f005:**
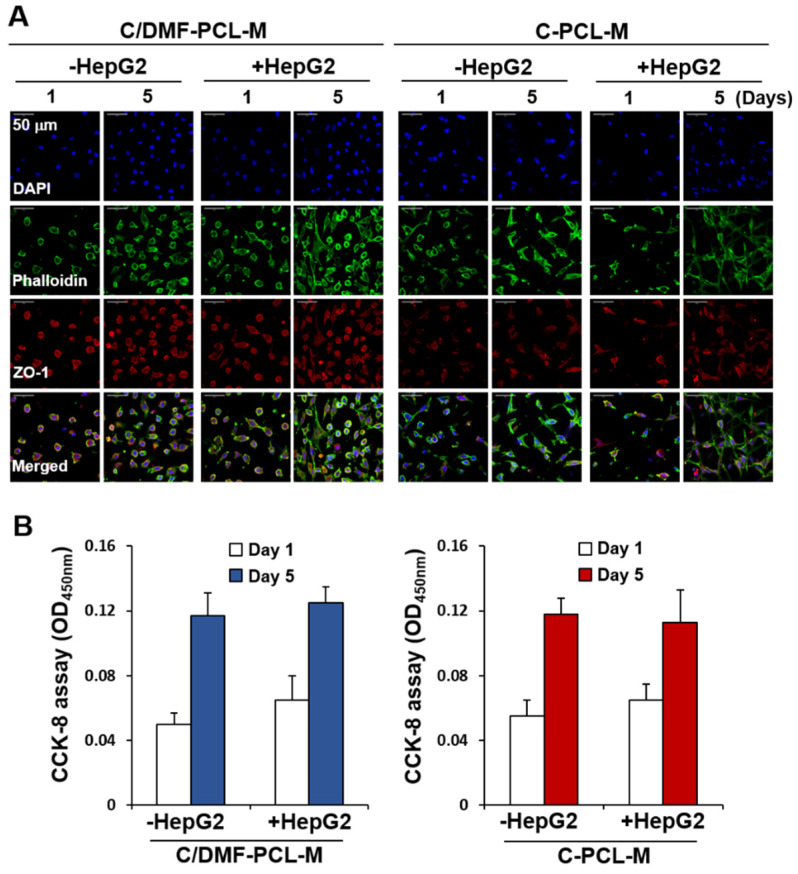
Growth of EA.hy926 ECs co-cultured with HepG2 cancer cells on C/DMF-PCL-M and C-PCL-M in a two-layer culture system. EA.hy926 cells (3 × 10^4^) were top-seeded on either C/DMF-PCL-M or C-PCL-M in the upper layer and cultured with (+HepG2) or without (−HepG2) HepG2 cells (3 × 10^4^) on the C-PCL-M in the lower layer. Cells were cultured for the indicated periods (*n* = 3). (**A**) Cells were stained with antibodies and cell distribution and morphology were assessed via confocal microscopy. Images represent three fields in each nanofibrous membrane. (**B**) Cell proliferation rates were analyzed using the CCK-8 assay. Data are presented as mean ± SD values (*n* = 3).

**Figure 6 ijms-21-04128-f006:**
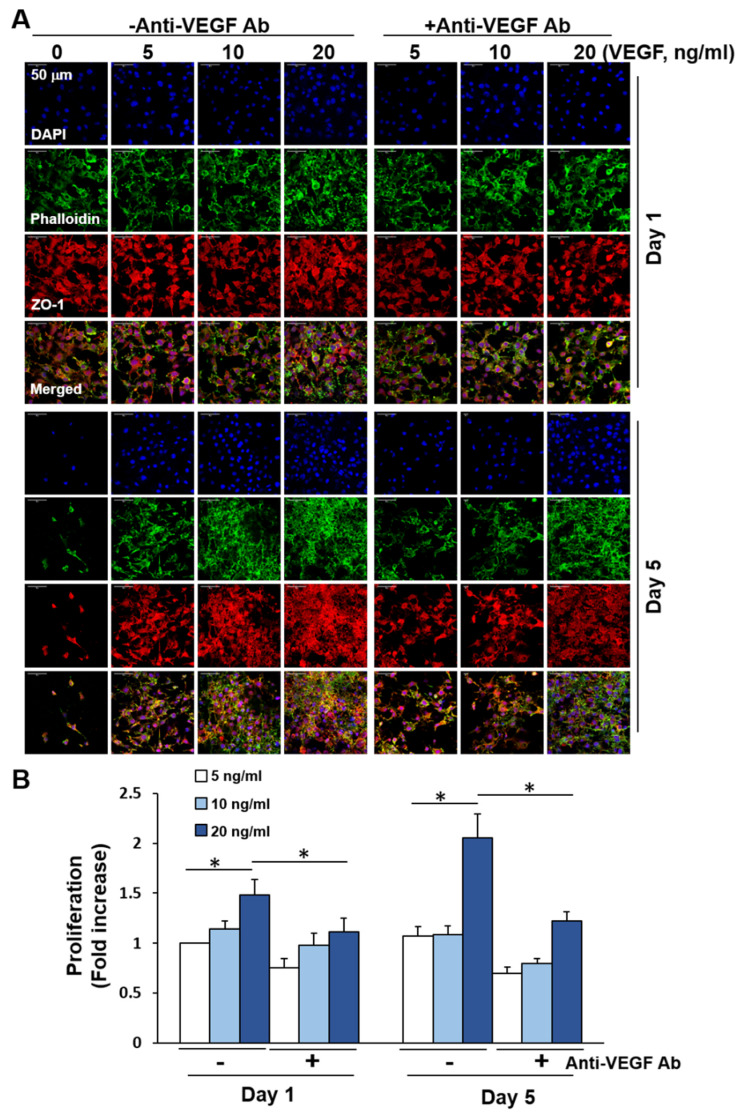
Growth regulation of bEND.3 ECs cultured on C/DMF-PCL-M by VEGF. bEND.3 cells (3 × 10^4^) were cultured with VEGF at increasing concentrations in the presence (+Anti-VEGF Ab) or absence (−Anti-VEGF Ab) of anti-VEGF antibody (0.6 μg/mL) for 1 d and 5 d (*n* = 3). (**A**) Cell morphology was assessed via confocal microscopy. Images represent three fields in each membrane. (**B**) The cell proliferation rate was analyzed using the CCK-8 assay and presented as a fold increase. Data are presented as mean ± SD values (*n* = 3). * *p* < 0.05.

**Figure 7 ijms-21-04128-f007:**
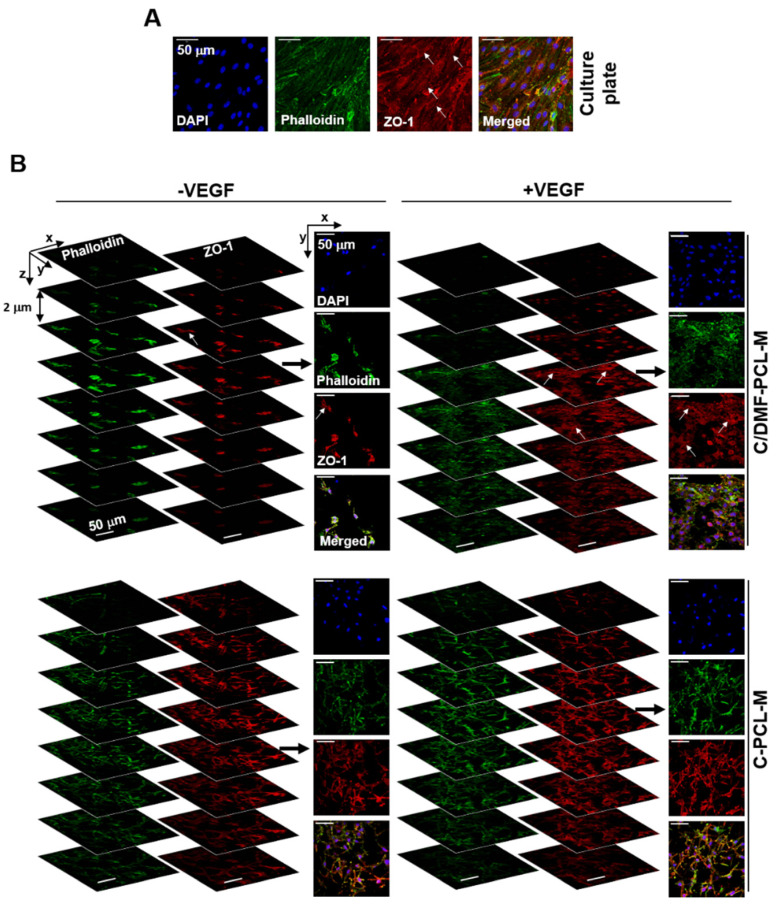
ZO-1 expression in bEND.3 ECs cultured on a culture plate and nanofibrous membrane scaffold. bEND.3 ECs (3 × 10^4^) were cultured for 5 d on a culture plate (**A**) and C/DMF-PCL-M or C-PCL-M in the presence (+VEGF) or absence (−VEGF) of VEGF (**B**) (*n* = 3). The formation of tight junctions between the cells (arrows) was evaluated using confocal microscopy after the cells were stained using anti-ZO-1 antibody (red). In panel B, the localization of tight functions was evaluated by focus-stacking images of ZO-1 expression in the cells. The data represent three independent experiments.

**Figure 8 ijms-21-04128-f008:**
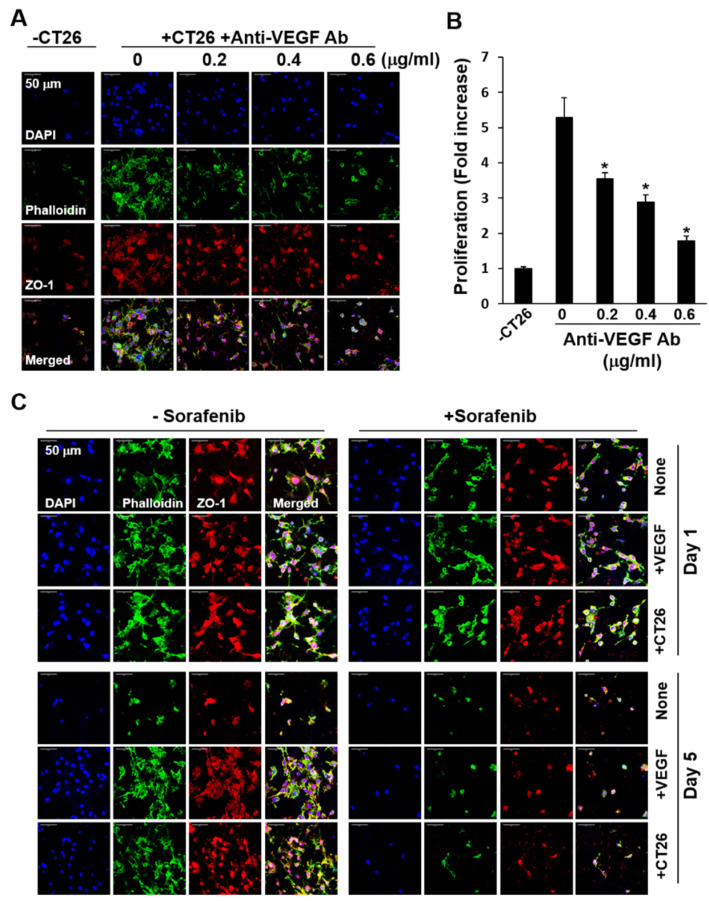
Inhibition of co-culture-induced growth of bEND.3 ECs by anti-VEGF antibody and sorafenib in a two-layer system. bEND.3 and CT26 cells (3 × 10^4^) were co-cultured in a two-layer system with anti-VEGF antibody at increasing concentrations for 5 d (*n* = 3). (**A**) bEND.3 cells on C/DMF-PCL-M in the upper layer were stained with DAPI, phalloidin, and anti-ZO-1 antibodies. Cell morphology was analyzed via confocal microscopy. (**B**) The proliferation of bEND.3 cells was assessed using the CCK-8 assay and presented as a fold increase. (**C**) bEND.3 cells were cultured alone (None), cultured with VEGF (+VEGF), and co-cultured with CT26 cells (+CT26) in the presence (+Sorafenib) or absence (−Sorafenib) of sorafenib (1 μM) for 1 d and 5 d (*n* = 3). Images represent three independent experiments. Data are presented as mean ± SD values (*n* = 3). * *p* < 0.05, compared to cultures without anti-VEGF Ab.

**Figure 9 ijms-21-04128-f009:**
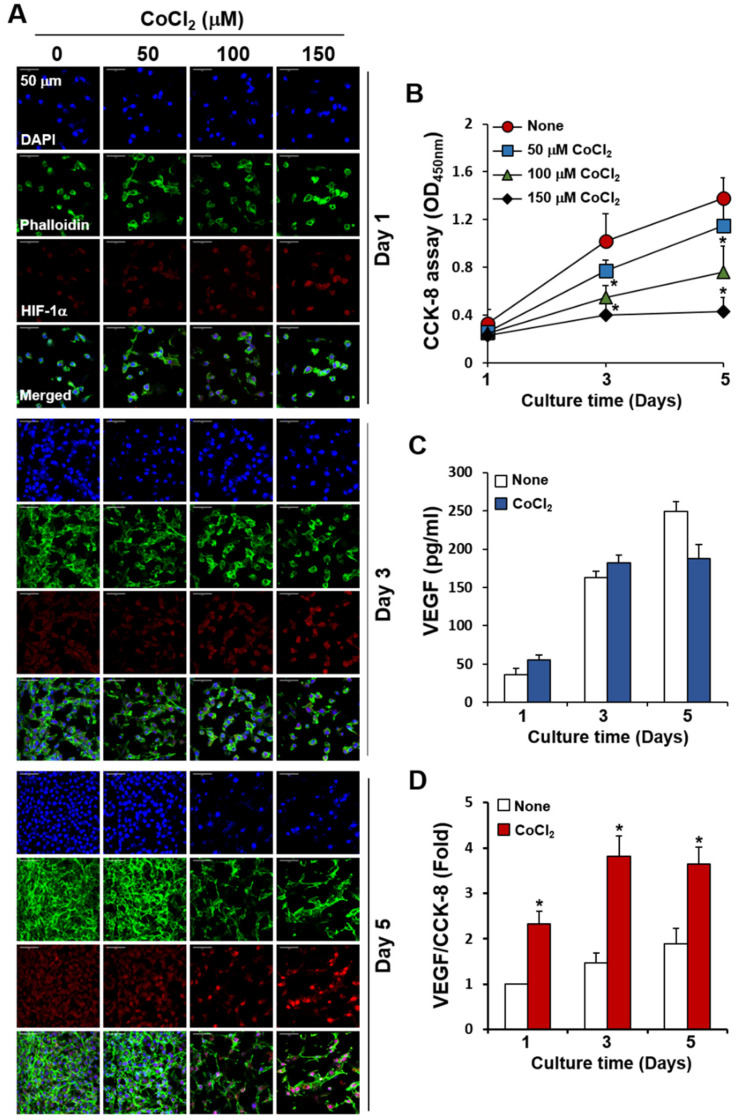
Decreased proliferation of CT26 cancer cells cultured on C-PCL-M and treated with CoCl_2_ at high concentrations. CT26 cells (3 × 10^4^) alone were top-seeded on C-PCL-M and cultured for the indicated periods with CoCl_2_ at increasing concentrations (*n* = 3). (**A**) Cells were stained with DAPI, phalloidin (green), and anti-HIF-1α antibodies (red). Cell distributions and fluorescence intensities were analyzed via confocal microscopy. (**B**) Proliferation rates of the cells were analyzed via the CCK-8 assay. (**C**) CT26 cell were treated with 150 μM CoCl_2_ and VEGF concentrations in the conditioned media were measured via ELISA. (**D**) CT26 cells were treated as in the panel C and mean VEGF concentrations (*n* = 3) were compared to mean CCK-8 levels (*n* = 3). Images represent three independent experiments. Data are presented as mean ± SD values (*n* = 3). * *p <* 0.05, compare to cultures without CoCl_2_.

**Figure 10 ijms-21-04128-f010:**
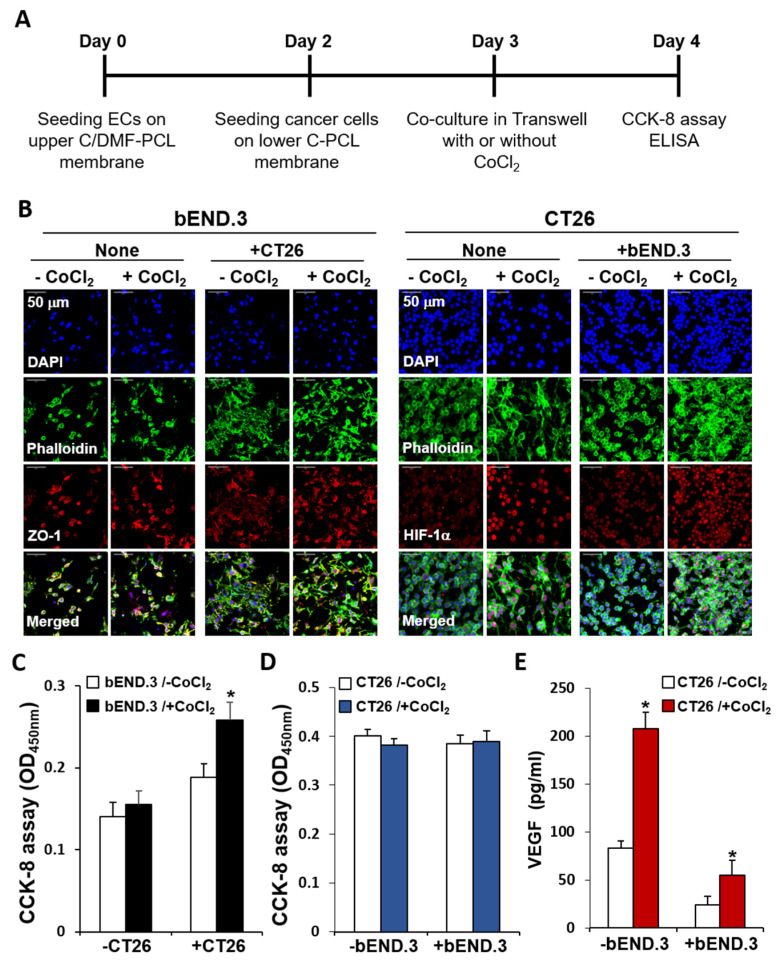
Increased growth of ECs by CoCl_2_ in the co-culture. (**A**) bEND.3 and CT26 cells (3 × 10^4^) were cultured alone or co-cultured (*n* = 3) as shown in the workflow diagram, to investigate the effects of CoCl_2_ on the growth of bEND.3 cells in a modified two-layer co-culture system. (**B**) The distribution and morphology of fluorescent antibody-labeled cells on the upper (bEND.3) and lower layers (CT26) were analyzed via confocal microscopy. (**C**) The proliferation rate of bEND.3 cells was determined via the CCK-8 assay after the ECs were cultured with (+CoCl_2_) or without (−CoCl_2_) CoCl_2_ in the presence (+CT26) or absence (+CT26) of CT26 cells. (**D**) The proliferation rate of CT26 cells was determined after cancer cells were cultured with (+CoCl_2_) or without (−CoCl_2_) CoCl_2_ in the presence (+bEND.3) or absence (-bEND.3) of bEND.3 cells. (**E**) CT26 cancer cells were cultured as in panel D and VEGF concentrations were determined via ELISA. Images represent three independent experiments. Data are presented as mean ± SD values (*n* = 3). * *p* < 0.05, compared to cultures without CoCl_2_.

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
