# Peer review of "Co-Culturing of Endothelial and Cancer Cells in a Nanofibrous Scaffold-Based Two-Layer System"

_ijms, 2020, doi:10.3390/ijms21114128_

Round 1
Reviewer 1 Report
In my opinion, the manuscript is now ready for publications.
Thanks for considering my suggestions :-)
Author Response
We thank the reviewer for his recommendation to publish the manuscript.
Reviewer 2 Report
In this manuscript, Oh and Choi et al. developed a 3D two-layer co-culture system for studying the effects of selected cancer cells on specific endothelial cells. Modeling tumor-driven angiogenesis is of great importance and broad interest. The authors’ effort in developing and examining this 3D system is much appreciated. In general, the experiments were well designed and the manuscript was well written. I do not have major concerns. Only a few minor suggestions:
- The authors have chosen two ECs for this study, bEND.3 mouse ECs and EA.hy926 human ECs. The reasoning for choosing these two ECs should be explained. Why they are more representative for this co-culture system? Do they respond differently to the VEGF pathway?
- Same suggestion also apply to the selected cancer cells. Why CT26 cancer cells (mouse) and HepG2 cells (human) are more representative? The authors need to explain the reasoning behind so the readers can judge whether it is possible to use this system for their own study.
- Page 2, from line 50. “When the release of angiogenic factors by cancer cells is very low and the proliferation rate of reactive ECs is very high, EC growth on co-culturing with cancer cells cannot be easily detected using an in vitro culture system.” Is the ECs proliferation rate one of the criteria to choose suitable cells for this co-culture system? Could the author add reference(s) here?
- Legend of Figure 3. I assume the morphology data is results represent four or five independent experiments? The CKK-8 assay is mean ± SD values (n = 3)? Please write clearly.
- In figure 5. The growth of EA.hy926 ECs was not increased by co-culture with HepG2 cells on CD-PCL-M and C-PCL-M. However, supplementary Figure S2 shows an increased growth on CD-PCL-M after short-term treatment with VEGF. Could the author show that VEGF was secreted by HepG2 during the co-culture?
Author Response
Reviewer-2
Comments and Suggestions for Authors
In this manuscript, Oh and Choi et al. developed a 3D two-layer co-culture system for studying the effects of selected cancer cells on specific endothelial cells. Modeling tumor-driven angiogenesis is of great importance and broad interest. The authors’ effort in developing and examining this 3D system is much appreciated. In general, the experiments were well designed and the manuscript was well written. I do not have major concerns. Only a few minor suggestions:
1. The authors have chosen two ECs for this study, bEND.3 mouse ECs and EA.hy926 human ECs. The reasoning for choosing these two ECs should be explained. Why they are more representative for this co-culture system? Do they respond differently to the VEGF pathway?
Answer) We thank the reviewer for the kind remarks.
As per the reviewer’s suggestions, we added the following text:
On page 6, line 11-14; The cells exhibiting the morphological, phenotypic, and functional characteristics of mouse and human ECs were chosen for our study and have been used for studying the EC migration and formation of capillary-like tubules [39,40].
Answer) Both bEND.3 mouse ECs and EA.hy926 human ECs express VEGFR-1 and -2. We added the following sentences to the text:
On page 9, line 5-6; bEND.3 cells are known to express VEGF receptor-1 and VEGF receptor-2 [47].
On page 9, line 15-16; Reports show that EA.hy926 cells express both VEGF receptor-1 and VEGF receptor-2 [48,49].
Cited references have been included and the order of references and the numbers updated.
2. Same suggestion also apply to the selected cancer cells. Why CT26 cancer cells (mouse) and HepG2 cells (human) are more representative? The authors need to explain the reasoning behind so the readers can judge whether it is possible to use this system for their own study.
Answer) We thank the reviewer for bringing this to our attention.
We added the following sentences as per the reviewer’s suggestion:
On page 7, line 14-17; Hypervasculature involvement in metastatic colorectal cancer and hepatocellular carcinoma development was shown earlier [43,44]. We used CT26 murine colon cancer and HepG2 human hepatocellular cancer cells since they secrete VEGF and induce angiogenesis [45,46].
3. Page 2, from line 50. “When the release of angiogenic factors by cancer cells is very low and the proliferation rate of reactive ECs is very high, EC growth on co-culturing with cancer cells cannot be easily detected using an in vitro culture system.” Is the ECs proliferation rate one of the criteria to choose suitable cells for this co-culture system? Could the author add reference(s) here?
Answer) As the reviewer alluded to, EC proliferation rate might be one of the criterions to choose EC type for determining the EC growth by cancer cells in a co-culture system. However, what we intended to convey was “ The EC growth on co-culturing with cancer cells cannot be detected readily when the levels of angiogenic factors released by cancer cells are insufficient to induce proliferation of ECs.” We believe that the original sentence is non-specific and therefore have removed it from the introduction section.
4. Legend of Figure 3. I assume the morphology data is results represent four or five independent experiments? The CKK-8 assay is mean ± SD values (n = 3)? Please write clearly.
Answer) We changed the sentence as follows:
On page 33, line 21; Data represent three independent experiments.
5. In figure 5. The growth of EA.hy926 ECs was not increased by co-culture with HepG2 cells on CD-PCL-M and C-PCL-M. However, supplementary Figure S2 shows an increased growth on CD-PCL-M after short-term treatment with VEGF. Could the author show that VEGF was secreted by HepG2 during the co-culture?
Answer) As suggested by the reviewer, we have included data on the amount of VEGF secreted by HepG2 cells in the Supplementary Figure 4B, and VEGF production by CT26 cells is shown in Supplementary Figure 4.
On page 10, line 8-11; HepG2 cells cultured on C-PCL-M secreted more VEGF than CT26 cells (Supplementary Figure S4B), although co-culture of HepG2 cells failed to increase EA.hy926 cell growth compared to the culture of EA.hy926 cell alone.
On page 23, line 16-19; Supplementary Figure S4. VEGF production by CT26 and HepG2 cells cultured on the nanofibrous membrane. CT26 (A) and HepG2 (B) cells (3 × 104) were cultured for the indicated periods on an 8-well culture plate Culture plate) and C-PCL-M attached to an 8-well plate (C-PCL-M).
Reviewer 3 Report
This paper is concerned with developing a PCL-nanofiber-based two-layer system for co-culturing endothelial cells and cancer cells. The results show that ECs could steady proliferate on PCL-M, and could be used as an angiogenic model for further analysis. There are some questions should be addressed clearly.
- English should be improved throughout the manuscript. For instance, in line 481, the authors should capitalize the first word in the sentence.
- There were various polymers have been used to enhance EC proliferation, why did the authors choose PCL? The authors should introduce more prior works, and point out the novelty in this paper in the revised version.
- Since the authors mentioned that the choice of solvent could influence nanofiber properties, it should be explained why the authors choose chloroform and dimethylformamide?
- How many times of the experiment were performed for each data that were presented in this article? Please clarify.
- I suggest the authors put more details in the methods section.
- In the manuscript, the abbreviation of electrospun PCL nanofibrous membranes is PCL-M, whereas in the figures, the abbreviation is different. Please keep everything constant.
- Why did the authors culture cancer cells on PCL-M? What’s the influences or benefits?
Author Response
Reviewer-3
Comments and Suggestions for Authors
This paper is concerned with developing a PCL-nanofiber-based two-layer system for co-culturing endothelial cells and cancer cells. The results show that ECs could steady proliferate on PCL-M, and could be used as an angiogenic model for further analysis. There are some questions should be addressed clearly.
1. English should be improved throughout the manuscript. For instance, in line 481, the authors should capitalize the first word in the sentence.
Answer) We thank the reviewer for bringing this to our attention, and the manuscript was checked for improving the English language.
On page page 20, line 1-3; In the two-layer system, bEND.3 cells were co-cultured with CT26 cells and EA.hy926 cells were co-cultured with HepG2 cells at 37oC in a humidified CO2 atmosphere, respectively.
2. There were various polymers have been used to enhance EC proliferation, why did the authors choose PCL? The authors should introduce more prior works, and point out the novelty in this paper in the revised version.
Answer) Following the reviewer’s suggestion, we included the following text to introduce the advantages of PCL use in this study:
On page 4, line 10-13; The biocompatibility and biodegradability of PCL polymers have been demonstrated in biological applications [26,27]. Earlier reports showed that vascular graft of PCL-M induces endothelialization and ECM formation, accompanied by degradation of implanted PCL nanofibers [27].
We included the cited references and corrected the order of reference numbers.
3. Since the authors mentioned that the choice of solvent could influence nanofiber properties, it should be explained why the authors choose chloroform and dimethylformamide?
Answer) We added the following sentences as per the reviewer’s suggestion:
On page 4, line 20-22; For a particular polymer-solvent system, the choice of solvent influences the formation of PCL nanofibers significantly and is one of the factors influencing the production of various fibers and potentially influencing nanofiber properties [31,32].
On page 4, line 22-page 5, line 2; Chloroform is the most commonly employed solvent for PCL electrospinning due to PCL solubility in chloroform [33]. N,N‐Dimethylformamide (DMF) due to its higher electric conductivity and lower vapor pressure, was used as a solvent additive to generate PCL nanofibers [34,35]. It was shown earlier that the electrospun PCL nanofiber diameter decreases with an increasing proportion of DMF in dichloromethane and DMF mixed solvents [35].
Cited references were included and the order of reference numbers corrected.
4. How many times of the experiment were performed for each data that were presented in this article? Please clarify.
Answer) We have included the numbers of experiments.
5. I suggest the authors put more details in the methods section.
Answer) As the reviewer suggested, some parts in the methods section were modified to include more detail.
On page 18, line 6; Each nozzle had an inner diameter of 210 μm (27G).
On page 19, line 5-7; PDMS (100 μl) was poured in 8-well plates and kept on a slide warmer at 100°C for 3 min. The PCL-M (1.1 cm × 1.3 cm) was attached to the surface of gel-state PDMS on 8-well plates and immersed in DMEM (700 μl) over 6 h at 37°C to increase cell adhesion.
On page 19, line 15-18; To generate the upper layer in the apical chamber, the polycarbonate filter of a Transwell insert was removed and replaced with a 5-mm-diameter circular section of C/DMF-PCL-M and C-PCL-M, which were attached with gel-state PDMS (40 μl).
On page 19, line 22-24; EC and cancer cell suspensions (10 μl) were seeded at 30,000 cells/well on the membranes in the upper and lower layers, respectively.
On page 19, line 24-25; The two layers were then assembled layer-by-layer by placing the insert in the well.
On page 20, line 1-3; In the two-layer system, bEND.3 cells were co-cultured with CT26 cells and EA.hy926 cells were co-cultured with HepG2 cells at 37oC in a humidified CO2 atmosphere, respectively.
6. In the manuscript, the abbreviation of electrospun PCL nanofibrous membranes is PCL-M, whereas in the figures, the abbreviation is different. Please keep everything constant.
Answer) Following the reviewer 4 suggestion, we changed CD-PCL-M to C/DMF-PCL-M and the abbreviations of C-PCL and CD-PCL have been changed to C-PCL-M and C/DMF-PCL-M in all figures, respectively.
7. Why did the authors culture cancer cells on PCL-M? What’s the influences or benefits?
Answer) Figure 3B shows that cancer cell proliferation is higher in C-PCL-M than on C/DMF-PCL-M. We added possible benefits as described below:
On page 7, line 19-22; Thus, in this study, cancer cells were cultured on C-PCL-M and not on C/DMF-PCL-M. In addition, the large pores in the C-PCL-M might facilitate deeper infiltration of cells into the scaffold and help maintain cell aggregates in the scaffold [28].
Reviewer 4 Report
Overall, this paper is very interesting and an extension of co-culture methods incorporating biomaterials in the culture system.
Minor revisions:
several font formatting/color issues (sometimes text appears blue or underlined)
C-PCL-M and CD-PCL-M are difficult to distinguish. If there is any other alternative way to differentiate between the two, that will make it easier to follow.
There is a reference in the abstract (line 16) to PCL-M. Which one is that?
Why is nanofibers (line84) a hyperlink?
The sample size for fiber diameter and pore size is unclear. Does n=20 in Figure 1 refer to the 20 random measurements or the number of samples analyzed (at least 5 samples per group)?
Why was ZO-1 chosen as the EC marker?
Please clarify line 130, 144: Data represent four or five independent experiments. Is it four or five?
Please clarify line 176, 187, 220: Images represent three or four fields in each membrane. Is it three or four? If one condition is three and a different condition is four, please specify.
In Figure 6B, the statistics bar is comparing the 3 concentrations of VEGF in absence of VEGF Ab, but the statistic bar comparing in presence of VEGF Ab is comparing 20ng/mL VEGF in absence of VEGF Ab and all three concentrations of VEGF in presence of VEGF Ab. Is this a mistake? If not, why this comparison?
Please clarify line 254 289, 329: Images represent three or four independent experiments. Is it three or four? Which condition is n=3 vs. n=4?
Why is the word nanofiber and cell adhesion hyperlinks? (lines395, 396)
Author Response
Reviewer-4
Comments and Suggestions for Authors
Overall, this paper is very interesting and an extension of co-culture methods incorporating biomaterials in the culture system.
Minor revisions:
1. several font formatting/color issues (sometimes text appears blue or underlined).
Answer) The blue color was used to help distinguish the revised and new sentences in the resubmitted manuscript from the first submitted manuscript. In addition, the manuscript has been checked for the English language.
2. C-PCL-M and CD-PCL-M are difficult to distinguish. If there is any other alternative way to differentiate between the two, that will make it easier to follow.
Answer) We thank the reviewer for bringing this to our attention, and we changed CD-PCL-M to C/DMF-PCL-M throughout the manuscript, including figures.
3. There is a reference in the abstract (line 16) to PCL-M. Which one is that?
Answer) We included a reference to the same sentence in the results section.
On page 5, line 12-13; PCL nanofibrous membranes were generated via electrospinning of PCL in chloroform (C-PCL-M) and chloroform and dimethylformamide (C/DMF-PCL-M) [28].
Cited reference has been included and the order of reference numbers corrected.
4. Why is nanofibers (line84) a hyperlink?
Answer) We fixed the typographical error in the MS word.
5. The sample size for fiber diameter and pore size is unclear. Does n=20 in Figure 1 refer to the 20 random measurements or the number of samples analyzed (at least 5 samples per group)?
Answer) To avoid confusion, we have changed the sentence as follows:
On page 18, line 12-14; Fiber diameter and pore size were measured by an average of 20 random measurements from SEM images of five membranes using the ImageJ software (ImageJ, National Institutes of Health, Bethesda, MD, USA).
6. Why was ZO-1 chosen as the EC marker?
Answer) In response to the reviewer’s question, we included the following sentence:
On page 6, line 18-20; The tight junction adaptor protein zona occludin (ZO)-1 is essential for barrier formation in microvascular EC and regulates the migration and angiogenic potential of ECs [41].
7. Please clarify line 130, 144: Data represent four or five independent experiments. Is it four or five?
Answer-1) As per the reviewer’s comment, we have corrected the number of experiments as follows:
On Figure 2 legend; Data represent four independent experiments.
Answer-2) Line 144 has been corrected as follows:
On Figure 3 legend; Data represent three independent experiments.
8. Please clarify line 176, 187, 220: Images represent three or four fields in each membrane. Is it three or four? If one condition is three and a different condition is four, please specify.
Answer) Following the reviewer’s suggestion, we updated the number of experiments as follows:
On Figure 4A legend; Images represent three fields in each membrane.
On Figure 5B legend; Images represent three fields in each nanofibrous membrane.
On Figure 6A legend; Images represent three fields in each membranes.
8. In Figure 6B, the statistics bar is comparing the 3 concentrations of VEGF in absence of VEGF Ab, but the statistic bar comparing in presence of VEGF Ab is comparing 20ng/mL VEGF in absence of VEGF Ab and all three concentrations of VEGF in presence of VEGF Ab. Is this a mistake? If not, why this comparison?
Answer) We thank the reviewer for bringing this to our attention, and we have modified sentences to explain the significance of the comparison shown in Figure 6A and B as follows:
On page 9, line 9-11; The cell proliferation assay revealed an increased proliferation of bEND.3 cells by VEGF (Figure 6B). Moreover, the VEGF-induced increase in EC growth and proliferation was blocked significantly upon treatment with anti-VEGF antibodies.
10. Please clarify line 254 289, 329: Images represent three or four independent experiments. Is it three or four? Which condition is n=3 vs. n=4?
Answer) As the reviewer pointed out, we have corrected numbers of experiments as follows:
On Figure 8 legend; Images represent three independent experiments.
On Figure 9 legend; Images represent three independent experiments.
On Figure 6A legend; Images represent three independent experiments.
11. Why is the word nanofiber and cell adhesion hyperlinks? (lines395, 396)
Answer) We fixed the typographical error in the MS word.
Round 2
Reviewer 3 Report
The manuscript could be accepted after minor revision. I suggest the authors keep the abbreviation constant in the manuscript and figures.
This manuscript is a resubmission of an earlier submission. The following is a list of the peer review reports and author responses from that submission.
Round 1
Reviewer 1 Report
The reviewed manuscript presents a very interesting study analyzing the interactions between endothelial and cancer cells in a 3D transwell system. The manuscript contains extensive and very properly done experiments. I highly recommend to publish this work, but not in the current form.
Minor points:
- Please change the title – it contains lots of repetitions. Further, a shorter title will be better, such as: „Co-culturing of endothelial and cancer cells in a nanofibrous scaffold-based two-layer system“
- At the beginning of the manuscript, data from human endothelial and cancer cells were shown however, the majority oft he experiments were performed with the murine cells. Please skip the experiments with the human cells, otherwise it gives the impression data are not complete. In the case that you skip this part , you can enlarge the other figures, for example figure 4
- Please correct all missing units in the whole manuscript
- Introduction: line 52 – release instead of amount; line 53 – low instead of small; line 56/57– in co-culture with cancer cells
- Please remove repeated figure numbers at the bottom right of the respective figure
- Section 2.2: line 111 – seeding instead of culturing; line 113 – exhibiting green and red fluorescence; line 114: in comparison to instead of with; line 114-115 – please remove because this description is not confirmed by the figure:“…, and the density of fluorescently stained cells increased after 3 d of culturing“.
- Figure 2: why do you choose 2 different magnifications?; Figure legend: Different growth pattern of ECs cultured on…(remove between)
- Page 7, line 167: „…, it was significantly increased upon co-culturing…“ – please give this significance also in Fig. 4D; line 178: as shown in the supplementary figure S1; please choose for Fig. 4D the same maximum fort he y-achsis
- Page 10, line 230: but EC growth was significantly inhibited; line 232: VEGF is the most potent factor; line 237: remove the space after HIF-1; line 254: CoCl2-treated cancer cells – which concentration of CoCl2 was used?
- Page 12: please add CoCl2 to the figure legend oft he diagram illustrated in fig. 8B; which CoCl2 was used for data illustrated in fig. 8C and D, please specify; line 286 - compared
- It’s not clear whether data illustrated in fig. 8 were generated in co-culture or not. I guess not, but please specify.
Author Response
Reviewer-1
Comments and Suggestions for Authors
The reviewed manuscript presents a very interesting study analyzing the interactions between endothelial and cancer cells in a 3D transwell system. The manuscript contains extensive and very properly done experiments. I highly recommend to publish this work, but not in the current form.
Minor points:
1. Please change the title – it contains lots of repetitions. Further, a shorter title will be better, such as: „Co-culturing of endothelial and cancer cells in a nanofibrous scaffold-based two-layer system“
Answer) As the reviewer suggested, we have changed the title to “Co-culturing of endothelial and cancer cells in a nanofibrous scaffold-based two-layer system”.
2. At the beginning of the manuscript, data from human endothelial and cancer cells were shown however, the majority of the experiments were performed with the murine cells. Please skip the experiments with the human cells, otherwise it gives the impression data are not complete. In the case that you skip this part, you can enlarge the other figures, for example figure 4.
Answer-1) As the reviewer suggested, we have removed Figure 2, Supplementary Figure S2, and Figure 5, which show results from experiments with EA.hy926 human endothelial cells.
Answer-2) In addition, all sentences describing results obtained with human cells (EA.hy926 and HepG2 cells) have been removed from the Abstract. Methods, Results, and Discussion sections of the manuscript.
Answer-3) Reference numbers have also been revised.
Answer-4) Figure 4 has been enlarged and shown as Figure 4-1 and 4-2 because it is bigger than one page.
3. Please correct all missing units in the whole manuscript.
Answer) We have cross-checked and corrected all missing units in the manuscript.
4. Introduction: line 52 – release instead of amount; line 53 – low instead of small; line 56/57– in co-culture with cancer cells
Answer) We have revised the sentences as follows: On page 3, line 23-24; When the release of angiogenic factors by cancer cells is very low and the proliferation rate of reactive ECs is very high, On page 4, line 1-3; Therefore, 3D co-cultivation is required to overcome the limitations including detection and cytotoxicity for long-term determination of the EC growth rate in co-culture with cancer cells.
5. Please remove repeated figure numbers at the bottom right of the respective figure
Answer) As the reviewer requested, we have removed the figure numbers in the revised manuscript.
6. Section 2.2: line 111 – seeding instead of culturing; line 113 – exhibiting green and red fluorescence; line 114: in comparison to instead of with; line 114-115 – please remove because this description is not confirmed by the figure:“…, and the density of fluorescently stained cells increased after 3 d of culturing“.
Answer) As the reviewer pointed out, the sentences have been corrected as follows: On page 6, line 5-8; As shown in Figure 2A, bEND.3 cells adhered to the nanofibers and were well-distributed throughout the scaffold in both nanofibrous membranes 1 d after seeding. The density of phalloidin- and ZO-1-labeled bEND.3 cells exhibiting green and red fluorescence in the CD-PCL-M significantly decreased 3 d after culturing. In comparison to CD-PCL-M, the growth of bEND.3 cells on C-PCL-M was stable.
7. Figure 2: why do you choose 2 different magnifications?; Figure legend: Different growth pattern of ECs cultured on…(remove between)
Answer-1) We have changed Figure 2 using the same magnification.
Answer-2) As suggested, the caption of Figure 2 legend was changed as follows: On page 28, line 9; Figure 2. Different growth patterns of bEND.3 ECs on the CD-PCL-M and C-PCL-M.
8. Page 7, line 167: „…, it was significantly increased upon co-culturing…“ – please give this significance also in Fig. 4D; line 178: as shown in the supplementary figure S1; please choose for Fig. 4D the same maximum for the y-achsis
Answer-1) As the reviewer suggested, we have added significance markers (*) in Figure 4D with the same maximum for the Y-axis and added the following annotation in the Figure 4 legend: *P < 0.05, compared to d 1
Answer-2) As the reviewer suggested, we have corrected the sentence as follows: On page 29, line 4 (Figure 4 legend); bEND.3 cells (3 × 104) were top-seeded on either CD-PCL-M (CD-PCL) or C-PCL-M (C-PCL) in the upper layer and cultured with (+CT26) or without (-CT26) CT26 cells (3 × 104) on the C-PCL-M in the lower layer as shown in the supplementary figure S1.
9. Page 10, line 230: but EC growth was significantly inhibited; line 232: VEGF is the most potent factor; line 237: remove the space after HIF-1; line 254: CoCl2-treated cancer cells – which concentration of CoCl2 was used?
Answer-1) As the reviewer pointed out, we have corrected the sentences as follows: On page 8, line 23; EC growth was significantly inhibited upon treatment with 1 μM sorafenib (Figure 6C). On page 9, line 1; VEGF is the most potent factor for inducing EC growth. On page 9, line 6; Effects of CoCl2 on cell growth, HIF-1α expression, and VEGF production in cancer cells cultured on C-PCL-M.
Answer-2) We have added the concentration of CoCl2 in the Figure 7 legend as follows: On page 30, line 14-17; (C) CT26 cells were treated with 150 μM CoCl2 and VEGF concentrations in the conditioned media were measured via ELISA. (D) CT26 cells were treated as in panel C and mean VEGF concentrations (n = 3) were compared to mean CCK-8 levels (n = 3).
10. Page 12: please add CoCl2 to the figure legend of the diagram illustrated in fig. 8B; which CoCl2 was used for data illustrated in fig. 8C and D, please specify; line 286 – compared
Answer-1) CoCl2 were shown in the diagram in Figure 7B.
Answer-2) We have corrected the Figure 7C and D legends by adding the concentrations of CoCl2 as in answer 9.
11. It’s not clear whether data illustrated in fig. 8 were generated in co-culture or not. I guess not, but please specify.
Answer) We have specified the monoculture of CT 26 as follows: On page 9, line 13-14; By comparing the cell distributions and densities of fluorescently labeled cells on the C-PCL-M layer, CT26 cells in monoculture condition displayed a significant reduction in cell numbers and growth upon treatment with 100 μM CoCl2 for 3 d in comparison with untreated control cells (Figure 7A). On page 30, line 10 (Figure 7 legend); CT26 cells (3 × 104) alone were top-seeded on C-PCL-M and cultured for the indicated periods with CoCl2 at increasing concentrations (n = 3).
Reviewer 2 Report
1. I did not get the authors‘ concept, why the extracellular matrix-free PCL substrate should be a relevant model. Moreover, the endotheloid cells grow as single cells in the PCL-scaffolds, mostly free of any intercellular contacts. A tube- or vessel-like structure of coherent ECs would recapitulate angiogenesis much better.
2. In line 86/87: the authors write: The nanofibres in both membranes were randomly oriented and structurally resembled collagen without bead formation. What do the authors mean with “without beads”?
3. EA.hy926 lack several features of endothelial cells. Did the authors check for EC-characteristic surface markers, e.g. VEGFRs by flow cytometry?
4. I think it to be highly unlikely that the cells adhere to the “naked” PCL-nanofibers. I presume, that fibronectin which is either in the cell culture medium or produced by the endotheloid cells would bridge the contact between cells and PCL-nanofibers. Did the authors test whether the bEND.3 and EA.hy926 produce fibronectin in different amounts? Whether the different nanofibers adsorb fibronectin with different coating efficieny? This, along with the potentially different repertoire of cell adhesion receptors of the endotheloid cells would explain the different morphologies and actin arrangements. Why didn´t the authors carry out these experiments also with primary human umbilical vein endothelial cells (HUVECs)?
5. In the method section: paragraph 4.4: were the ECs and cancer cells only cultured in DMEM without fetal calf serum?
6. ZO-1 is a marker for tight cell junctions. However, the cells in the nanofiber scaffolds have barely intercellular contacts. What does the staining of endotheloid cells without cell-cell contacts mean?
7. Maybe I did not understand the protocol for cell counting in Fig. 4 and 5 correctly. The CCK-8 kit does not specifically stain endotheloid cells, nor tumor cells. Hence, have the two nanofiber scaffolds containing the endotheloid cells or cancer cells been taken apart before staining and cell-counting? Or how could the authors conclude cell type specificity in the CCK-8 assay?
8. The greek letters, e.g., epsilon (?)-captolacton, mikro-meter, and mikro-liters, have disappeared in the manuscript. Especially, if concentration values are affected from missing prefixes, the manuscript becomes hard to interpret.
Minor points:
1. The texts for funding and acknowledgement have to be swapped.
Author Response
Comments and Suggestions for Authors
1. I did not get the authors‘ concept, why the extracellular matrix-free PCL substrate should be a relevant model. Moreover, the endotheloid cells grow as single cells in the PCL-scaffolds, mostly free of any intercellular contacts. A tube- or vessel-like structure of coherent ECs would recapitulate angiogenesis much better.
Answer-1) As the reviewer pointed out, normal ECs in tissue grow in a coherent pattern and form tube-like structure. In this study, we tried to mimic EC growth in tumor tissue using nanofiber scaffold. The authors observed that ECs look like single cells without the formation of tight junction because the cultured ECs in the PCL scaffold were oriented three dimensionally and the contact surface between adjacent ECs were very small compared to the two dimensional culture condition.
Thus, we added following sentence in the Discussion. On page 13, line 15-18; ECs in CD-PCL scaffold look like single cells without the formation of tight junction because the ECs in the PCL nanofiber scaffold were oriented three dimensionally and the contact surface between adjacent ECs, which grew along the fibers, were very small compared to 2D culture.
Answer-2) In ongoing experiments with another nanofiber membrane using poly(vinyl alcohol) (PVA) nanofibers, we observed coherent growth of bEND.3 cells (shown in the figures for reviewer). Thus, cell growth pattern may be dependent on the pore sizes and characteristics of nanofibers. We are conducting co-culturing experiments with ECs on PVA nanofiber layer and cancer cells on PCL nanofiber layer, similar to PCL nanofiber- based two-layer system.
Answer-3) Compared to PCL nanofibers, PVA nanofibers have the disadvantage of lower cell adherence without blending of cell adhesion molecules, such as fibronectin and laminin. Thus, in this study, we used PCL nanofiber scaffold in the co-culture of ECs and cancer cells.
2. In line 86/87: the authors write: The nanofibres in both membranes were randomly oriented and structurally resembled collagen without bead formation. What do the authors mean with “without beads”?
Answer) As the reviewer pointed out, we have corrected the sentences as follows to clear SEM results: On page 5, line 11-13; The nanofibers in both membranes were randomly oriented and structurally resembled collagen (Figure 1A). The structure of electrospun nanofibers showed a uniform distribution without bead formation.
3. EA.hy926 lack several features of endothelial cells. Did the authors check for EC-characteristic surface markers, e.g. VEGFRs by flow cytometry?
Answer-1) As the reviewer pointed out, EA.hy926 cells are human umbilical vein cells established by fusing primary human umbilical vein cells with A549 cells. We obtained the cells from ATCC; cells in passage 6-10 were used in the experiments. It is known that EA.hy926 cells express VEGFR1 and VEGFR2 (D’Hane et al., Plos Ono, 2013:8(6):e67029; Nitzsch et al., Br. J. Cancer, 2010:103:18-28.
Answer-2) As reviewer 1 highly recommended, we have removed the results with human EA.hy926 ECs and cancer cells from this manuscript. Thus, we did not include the findings related to EA.hy926 cells.
4. I think it to be highly unlikely that the cells adhere to the “naked” PCL-nanofibers. I presume, that fibronectin which is either in the cell culture medium or produced by the endotheloid cells would bridge the contact between cells and PCL-nanofibers. Did the authors test whether the bEND.3 and EA.hy926 produce fibronectin in different amounts? Whether the different nanofibers adsorb fibronectin with different coating efficieny? This, along with the potentially different repertoire of cell adhesion receptors of the endotheloid cells would explain the different morphologies and actin arrangements. Why didn´t the authors carry out these experiments also with primary human umbilical vein endothelial cells (HUVECs)?
Answer-1) As the reviewer pointed out, we cannot rule out the possibility that cell binding ECM proteins can be produced and adsorbed to PCL nanofibers because PCL nanofiber has no cell-binding molecules, whereas cell growth, proliferation, and differentiation can be promoted in cultured cells on PCL nanofiber scaffold. In this study, Figure 2A shows that the adhesion of ECs in nanofibrous scaffold was not different between culturing ECs on the CD-PCL-M and C-PCL-M. We speculated the difference between spreading of cells on CD-PCL-M and C-PCL-M may be due to different fiber structures. We have described the results in the results 2.2 section. Instead of adherence, our results showed that ECs in CD-PCL-M rather than C-PCL were more responsive to added VEGF and co-culturing with cancers cells.
Answer-2) In the Discussion section, we mentioned the possibility of having differences in ECM protein and serum protein coating in cell adhesion and proliferation. Thus we added the sentences as follows: On page 15, line 3-5; Moreover, it is possible that cell binding ECM proteins, such as fibronectin can be produced by ECs and cancer cells and adsorbed to PCL nanofibers with different coating efficiency. Thus, differential adsorption of serum or ECM proteins in CD-PCL and C-PCL nanofibers may affect EC binding to the scaffolds.
Answer-3) We carried out the experiments with HUVEC in 3D culture, but the cell growth was not significantly increased in the presence or absence or VEGF (data not shown), whereas EA.hy926 cell growth maintained in the 3D condition. Thus, we have used EA.hy926 cells in human cell co-culturing model.
Answer-4) As an answer to reviewer 1’s suggestion, we have removed the results with human cells.
5. In the method section: paragraph 4.4: were the ECs and cancer cells only cultured in DMEM without fetal calf serum?
Answer) We have corrected the sentence as follows: On page 17, line 12-13; After 4 h of cell adhesion, ECs and cancer cells on the membranes were cultured in DMEM (700 μl) containing 10% FBS supplemented with penicillin/streptomycin for up to 5 d.
6. ZO-1 is a marker for tight cell junctions. However, the cells in the nanofiber scaffolds have barely intercellular contacts. What does the staining of endotheloid cells without cell-cell contacts mean?
Answer) ECs were stained with anti-ZO-1 antibody to determine whether tight junction is formed in the 3D culture of ECs. As the answer to reviewer 2’s suggestion in the No 1, ZO-1 expression was mainly observed in cytoplasm rather than on plasma membrane, indicating that tight junction formation is very low in 3D nanofibrous scaffold owing to the three dimensional orientation and low contact surface between ECs, which grow along the fibers.
7. Maybe I did not understand the protocol for cell counting in Fig. 4 and 5 correctly. The CCK-8 kit does not specifically stain endotheloid cells, nor tumor cells. Hence, have the two nanofiber scaffolds containing the endotheloid cells or cancer cells been taken apart before staining and cell-counting? Or how could the authors conclude cell type specificity in the CCK-8 assay?
Answer) We performed CCK-8 assay of the co-cultured cells in a two-layer system as follows: On page 19, line 9-12 (section 4.5); When ECs and cancer cells were co-cultured in a two-layer system using a Transwell chamber, the upper layer of the Transwell insert was separated from the lower chamber culturing cancer cells and replaced in a new chamber, and CCK-8 solution was then added to each well containing ECs or cancer cells.
8. The greek letters, e.g., epsilon (?)-captolacton, mikro-meter, and mikro-liters, have disappeared in the manuscript. Especially, if concentration values are affected from missing prefixes, the manuscript becomes hard to interpret.
Answer) The authors have cross-checked and corrected the missing units and Greek letters.
Minor points:
The texts for funding and acknowledgement have to be swapped.
Answer) As the reviewer recommended, we have included grant supports in the Funding section
Round 2
Reviewer 2 Report
The authors did NOT answer my questions adequately. Just by omitting the data of one cell line or by telling that endothelial cells do not form cell-cell-junction because they were grown in a 3D environment, is not sufficient to improve the manuscript.